# Plastics in the Indian Ocean – sources, transport, distribution and impacts

Charitha Pattiaratchi[1], Mirjam van der Mheen[1], Cathleen Schlundt[2], Bhavani E. Narayanaswamy[3], Appalanaidu Sura[4], Sara Hajbane[1], Rachel White[5], Nimit Kumar[6], Michelle Fernandes[7], Sarath Wijeratne[1]

[1]Oceans Graduate School and the UWA Oceans Institute, the University of Western Australia, Perth, 6009, Australia
[2]GEOMAR, Helmholtz Centre for Ocean Research, Kiel, Düsternbrooker Weg 20, 24105 Kiel, Germany
[3]Scottish Association for Marine Science, Oban, Argyll, PA37 1QA, Scotland, UK
[4]National Centre for Coastal Research, Ministry of Earth Sciences, Chennai, 600100, India
[5]School of Biological Sciences, the University of Western Australia, Perth, 6009, Australia
[6]Ministry of Earth Sciences (MoES), Indian National Centre for Ocean Information Services (INCOIS), Hyderabad-500090, India.
[7]National Centre for Polar and Ocean Research, Ministry of Earth Sciences, Goa, India

*Correspondence to*: Charitha Pattiaratchi (chari.pattiaratchi@uwa.edu.au)

**Abstract**

Plastic debris is the most common and exponentially increasing human pollutant in the world's ocean. The distribution and impact of plastic in the Pacific and Atlantic Oceans have been the subject of many publications but not so the Indian Ocean (IO). Some of the IO rim countries have the highest population densities globally and mismanagement of plastic waste is of concern in many of these rim states. Some of the most plastic-polluted rivers empty into the IO with all this suggesting that the IO receives a tremendous amount of plastic debris each year. However, the concentration, distribution and impacts of plastics in the IO are poorly understood as the region is under-sampled compared to other oceans. In this review, we discuss sources and sinks, which are specific to the IO. We also discuss unique atmospheric, oceanographic, and topographic features of the IO that control plastic distribution, such as: reversing wind directions due to the monsoon, fronts, and upwelling regions. We identify hotspots of possible plastic accumulation in the IO, which differ between the two hemispheres. In the southern IO, plastics accumulate in a garbage patch in the subtropical gyre. However, this garbage patch is not well defined, and plastics may leak into the southern Atlantic or the Pacific Ocean. There is no subtropical gyre and associated garbage in the northern IO due to the presence of landmasses. Instead, the majority of buoyant plastics most likely end up on coastlines. Finally, we identify the vast knowledge gaps concerning plastics in the IO and point to the most pressing topics for future investigation.

## 1. Introduction

Historically, the motivation for developing synthetic materials like plastics was for the conservation of elephants that inhabit countries along the Indian Ocean (IO) rim in southern Asia and Africa (Freinkel, 2011). The first plastic materials were advertised as saviours of the environment because it would no longer be necessary to ransack the environment for scarce natural resources (Meikle, 1997). However, the production of plastic materials has increased exponentially since the 1950s (PlasticsEurope, 2019) and plastics have instead become a ubiquitous environmental pollutant (Law, 2017). Since a large percentage of all plastics are single-use packaging items (PlasticsEurope, 2019), plastic waste has increased at a similar rate (Geyer et al., 2017). Since the 1950s, the global human population has generated an estimated 8300 million tonnes of plastic, of which over 75% (6300 million tonnes) has been discarded as waste (Geyer et al., 2017). Almost 80% of this plastic waste either ends up in landfill or in the environment. As a result, several million tonnes of plastic waste are estimated to enter the global oceans every year from coastlines (Jambeck et al., 2015), from inland sources transported by rivers (Lebreton et al., 2017; Schmidt et al., 2017; 2018); or directly from ocean-based sources such as the global fishing industry (Richardson et al., 2019b), offshore platforms and commercial and recreational shipping.

Plastic materials are durable and very slow to degrade and can persist in the marine environment for decades to centuries (Barnes et al., 2009). Around 35% of all plastic materials produced globally have densities higher than that of seawater (PlasticsEurope, 2019), and thus can sink to the seafloor. The remaining 65% of plastics float in the ocean and can travel enormous distances in the ocean. Photodegradation and other weathering processes at the sea surface lead to fragmentation, increasing micro- and nano- plastic abundance (Andrady, 2011). Plastics also accumulate biofouling while in the ocean, which can change the overall density and lead to plastics moving vertically in the water column (Lobelle & Cunliffe, 2011; Long et al., 2015; Kooi et al., 2017). As a result, plastics are ubiquitous throughout the marine environment and have been found on remote islands (Duhec et al., 2015; Lavers and Bond, 2017), in polar ice (Obbard et al., 2014; Bergmann et al., 2017; Peeken et al., 2018), and in the deep-seas (Van Cauwenberghe et al., 2013; Jamieson et al., 2019; Courtene-Jones et al., 2020).

Plastic pollution has several known harmful effects on marine species and ecosystems, including ingestion, entanglement, and the transport of potentially invasive species into foreign remote habitats (Gregory, 2009; Law, 2017). Plastics can also absorb and transfer toxicants (Rochman et al., 2012), potentially leading to the accumulation of toxins across the food web (Engler, 2012). Marine plastic debris also causes significant economic damage to industries and communities, with estimated costs ranging between USD 8 billion (Raynaud, 2014) to USD 2500 billion (Beaumont et al., 2019). Economic costs can be caused by damage to ships and fishing equipment when they collide with big marine plastic fragments (Richardson et al., 2019a) and by soiled landscapes that impair the tourism industry when it relies on clean beaches and coastal waters (Sari et al., 2021).

However, the overall impact of plastic pollution in the ocean is still poorly understood (Law, 2017). There are challenges in investigating ecological impacts and the fate of plastics in the ocean. Estimates of the amount of plastic floating on the global ocean surface are approximately 1% or less of the estimated amount of plastic that enters the ocean per year (van Sebille et al., 2015). This discrepancy in the global ocean plastic budget highlights a fundamental gap in the understanding of the fate of ocean plastic and suggests that there are unknown sinks of marine plastic debris. Possible sinks include biological sinks, i.e.

organisms ingesting plastics (e.g. Davison and Asch, 2011); sea ice cover temporarily accumulating plastics (Obbard et al., 2014; Peeken et al., 2018); fragmentation and biofouling of plastics leading to settling (e.g. Koelmans et al., 2017); and plastics making landfall ("beaching") and accumulating on the world's coastlines (e.g. Lebreton et al., 2019; van der Mheen et al., 2020c).

Plastic debris in the IO has been under-sampled and under-studied compared to other oceans, whilst the fate of plastics has not been investigated in any detail. However, a large percentage of all global ocean plastic is estimated to enter the IO: up to 15% of all coastal plastic (Jambeck et al., 2015) and 20% of all riverine plastic (Lebreton et al., 2017; Schmidt et al., 2017; 2018). Half of the top ten countries contributing most to ocean plastic pollution are located along the IO rim (Indonesia, Thailand, Malaysia, India and Bangladesh), and two of the largest and most polluting rivers (Ganges and Indus) empty into the IO. In

addition, it has been estimated that the IO contains the second largest plastic load in the ocean after the North Pacific Ocean (Eriksen et al., 2014), although there are insufficient measurements of plastics available to confirm this. In this synthesis paper, we compile existing knowledge about the sources (section 2), observations (section 3), transport and fate (section 4), impact (section 5) of plastic debris in the IO as well as highlight some of the emerging policies and initiatives (section 6), knowledge gaps and recommend future research strategies (section 8). We complete the paper by referring to a recent ship accident that

released 78,000 tonnes of nurdles (plastic pellets) into the Indian Ocean off Sri Lanka in May 2021 (section 7).

## 2. Sources

Plastic waste enters the ocean from coastal sources transported by wind and tides (Jambeck et al., 2015), from sources far into the hinterland transported by rivers (Lebreton et al., 2017; Schmidt et al., 2017; 2018), and directly from ocean-based sources (Richardson et al., 2019b). The IO is surrounded by 34 countries with population densities of around 100 people per $km^2$ on

average, with Australia the least populated country (3 people per $km^2$) and India the most populated country (400 people per $km^2$). Hoornweg and Bhada-Tata (2012) estimated that the total amount of plastic waste produced by all IO rim countries in 2010 was around 41 million tonnes. In comparison, the total amount of plastic waste produced in 2010 by the USA and China alone is estimated to be 38 and 59 million tonnes, respectively. More recently, Kaza et al. (2018) estimated that the total amount of plastic waste produced by IO rim countries in 2016 was around 24 million tonnes, compared to 34 million tonnes by the

USA and 39 million tonnes by China. Despite the relatively low plastic waste production around the IO rim, a large percentage

ends up in the environment because of poor waste management in most Asian and African countries. Jambeck et al. (2015) estimated that around 73% of plastic waste is inadequately managed along the IO rim and is released into the environment.

## 2.1 Land-based sources

Jambeck et al. (2015) estimated the global amount of plastic waste that entered the ocean in 2010 from coastal populations living within 50 km of the coastline. Based on a fixed percentage of mismanaged plastic waste entering the oceans (15% for the low-range estimates and 40% for the high-range estimates), they estimated that between 4.8 and 12.7 million tonnes of plastic entered the global oceans. However, it is likely that the estimated amount of plastic waste entering the ocean by Jambeck et al. (2015) for Sri Lanka is incorrect. Jambeck et al. (2015) based their estimate on a reported 5.1 kg of municipal solid waste generated per person per day in Sri Lanka (Hoornweg and Bhada-Tata, 2012). The updated report by Kaza et al. (2018) and dataset available through the World Bank ('What A Waste Global Database') indicate that only 0.34 kg of municipal solid waste is generated per person per day in Sri Lanka; this number is also more in line with the amount of waste generated in other developing countries. Using this correction, the amount of plastic waste entering the ocean from Sri Lanka through coastal populations is estimated to be between 0.021 and 0.057 million tonnes in 2010, instead of between 0.24 and 0.64 million tonnes as reported in Jambeck et al. (2015). Using this corrected number, around 15% of global ocean plastic entered the IO directly through coastal sources (Figure 1a; showing corrected waste input for Sri Lanka).

Lebreton et al. (2017) and Schmidt et al. (2017) estimated the amount of plastic waste entering the oceans through rivers. Similar to Jambeck et al. (2015), their estimates are based on a percentage of mismanaged plastic waste and include the influence of river catchment geography and river discharge. In addition, these estimates were calibrated based on available measurements of river plastic debris globally, ranging between sizes of 0.3 mm to 0.5 m. Lebreton et al. (2017) estimated that between 1.15 and 2.41 million tonnes of plastic waste enter the global ocean per year, up to 20% of which enters the IO (Figure 1b). Schmidt et al. (2017) estimated that between 0.47 and 2.75 million tonnes of plastic waste enter the global ocean per year, of which around 15% enters the IO. More recently, Meijer et al. (2021) estimated that between 0.80 and 2.7 million tonnes of macroplastics (defined by Meijer et al., 2021 as larger than 5 mm) enter the global ocean per year. In this estimate, Meijer et al. (2021) took into account the spatial variability of mismanaged plastic waste generated within a river basin, as well as more advanced climate and terrain characteristics than considered in the estimates of Lebreton et al. (2017) and Schmidt et al. (2017). They calibrated their estimates based on visual sampling of macro-plastics at river mouths around the world.

The estimates of the amount of plastic waste entering the oceans through rivers by Lebreton et al. (2017), Schmidt et al. (2017), and Meijer et al. (2021) agree relatively well with each other. Jambeck et al. (2015) estimated the amount of plastic waste entering the oceans through coasts are an order of magnitude higher. In even starker contrast, Weiss et al. (2021) re-evaluated the estimates of Lebreton et al. (2017) and Schmidt et al. (2017) and suggested that only 6.1 thousand tonnes of microplastics

(defined by Weiss et al., 2021 as smaller than 5 mm) enter the ocean through rivers each year, which is 2 to 3 orders of magnitude smaller than previous estimates. These differences highlight the extreme uncertainty involved in determining the amount of plastic waste entering the ocean from land-based sources. These estimates are based on few measurements of plastics entering the ocean (in the case of Jambeck et al., 2015, only on data from the San Francisco Bay; in the case of Lebreton et al., 2017; Schmidt et al., 2017; Meijer et al., 2021; and Weiss et al., 2021, on 30 to 340 samples from 13 to 89 rivers around the world). None of these samples was from IO rim countries or in rivers that empty into the IO. Expanding on these datasets will likely help improve these estimates, especially for the IO. However, as Weiss et al. (2021) demonstrate, to reduce extreme errors it is essential to use comparable sampling methodologies and to collect not only data on the amount of plastics sampled but also on, it is essential to use comparable sampling methodologies and collect data on the amount of plastics sampled and their weight. Furthermore, Meijer et al. (2021) emphasize the importance of sampling plastics at river mouths to get a more reliable estimate of the amount of plastic that actually enters the ocean. However, sampling plastics further upstream in addition to the river mouth, can help improve models of the probability for plastic to reach the ocean from inland areas.

Based on the currently available estimates, the largest coastal and riverine plastic sources in the IO are in the northern hemisphere around the Bay of Bengal and on the eastern side of the Arabian Sea (Figure 1). The input of plastic waste from rivers mainly depends on the river discharge, which varies seasonally. The highest river discharges occur during the wet season, during the boreal summer in the northern IO. Lebreton et al. (2017) estimated that plastic waste input from rivers in the IO peaks in August. In the southern hemisphere, the largest coastal and riverine sources of IO plastic waste are from Indonesia (Figure 1).

## 2.2   Ocean-based sources

Plastic waste can also enter the ocean directly from ocean-based sources such as the fishing industry, commercial and recreational shipping, and offshore platforms. In 1988, the International Convention for the Prevention of Pollution from Ships (MARPOL) prohibited waste dumping from vessels.  However, accidental losses and illegal dumping still contribute to plastic debris (Richardson et al., 2021; Stelfox et al., 2020a; see section 7 for a case study of a recent incident off the coast of Sri Lanka). Abandoned, Lost, or Discarded Fishing Gear (ALDFG) by the fishing industry can produce large quantities of plastic debris, i.e. monofilament lines and nets primarily made from synthetic materials (Sheavly and Register, 2007; Bond et al., 2012). Since 2000, around 13% of the global marine capture of fishes originated from the IO region (Pauly and Zeller, 2016). It is, therefore, likely that ALDFG is a significant source of plastic debris in the IO. Estimates of plastic waste entering the ocean from fishing vessels (e.g. ghost nets) or offshore platforms do not currently exist.

Finally, plastic waste originating from south-east Asia can also be transported to the IO by the Indonesian Throughflow (section 4.1). However, these sources are currently undocumented and need to be further investigated.

## 3. Observations

Plastics have been sampled in the IO in coastal waters, the open ocean, sediment, and organisms since the 1980s (Table 1), with most sampling studies being conducted on beaches. However, measurements are relatively scarce, and plastics have been regularly under-sampled in the IO (Figure 2) compared to the Atlantic and Pacific oceans. Samples of plastic debris consist of different plastic polymers and are generally classified into different types and size categories. Size and type categories can vary widely between authors but it is beyond the scope of this review to discuss these different categories. Instead, we refer to recent review papers by Gigault et al. (2018) and Frias & Nash (2019) discussing plastic size categories, and Hartmann et al. (2019) discussing different categories of polymers, sizes, shapes, colours, and origins.

Global open ocean plastic samples were standardised by van Sebille et al. (2015) and the plastic concentrations from these samples in the IO can be quantitatively compared (Figure 2a). In contrast, the methods used sampling plastics on beaches and in sediment vary widely (as illustrated in Table 1) and offer only a qualitative confirmation that plastics have been found on beaches and in sediment throughout the IO (Figure 2b). As discussed extensively in the review by Serra-Gonçalves et al. (2019), adopting a standardised framework to collect and report on beach debris is essential for these studies to be useful to the wider scientific community. Isobe et al. (2019) discuss the importance of a standardised protocol for laboratory analysis of plastics.

Multiple studies of plastic ingestion by different types of fauna have been done in the IO (Table 1, section 4.3.2 and section 5.2).

## 4. Transport and fate

Approximately 35% of all produced plastic materials have densities higher than that of seawater (PlasticsEurope, 2019) and, therefore will sink to the seafloor after entering the ocean. To the best of our knowledge, there have been no studies in the IO on sinking plastics and the transport of plastics through the water column and along the seafloor. Most plastic materials float in the ocean and are transported by ocean surface dynamics (currents, wind, and waves). However, buoyant plastic debris can be vertically distributed in the upper few metres of the water column because of wind- and wave- induced mixing (Kukulka et al., 2012; Kukulka and Brunner, 2015; Brunner et al., 2015). Depending on the sea state and the characteristics of the plastic debris, sampling studies in the North Atlantic Ocean found plastic debris mixed up to 5 m depth in the water column (Reisser et al., 2015; Kooi et al., 2016). In addition, the density of plastic debris can change under the influence of biofouling and degradation (Lobelle & Cunliffe, 2011; Long et al., 2015; Kooi et al., 2017). As a result, initially buoyant plastics may then sink (e.g. Koelmans et al., 2017). These plastics can also become de-fouled in the water column because of foraging and decreasing light intensities, leading to plastics oscillating in the water column (Andrady, 2011; Kooi et al., 2017). Fallouts of

plastic into the water column and the deep sea have recently been detected below the subtropical garbage patch in the North Pacific Ocean (Egger et al., 2020). In the IO, mixing of buoyant plastics in the water column and sinking of initially buoyant plastics have not yet been studied. This section focuses only on the transport of buoyant plastics by ocean surface dynamics in the IO.


Different forcing mechanisms transport buoyant plastics drifting at the ocean surface. Recent reviews by Zhang et al. (2017) and van Sebille et al. (2020) discuss these processes and their influence on the transport of plastics in detail. Ocean surface currents play a dominant role in the transport of buoyant objects, but wind (Breivik and Allen, 2008) and surface gravity waves (Röhrs, et al., 2012) can significantly influence an object's behaviour. Ocean surface currents are forced by many different

mechanisms such as wind, waves, tides, and density gradients (Talley et al., 2011; van Sebille et al., 2020). Combined with the Coriolis force, these forcing mechanisms result in Ekman currents, geostrophic currents, and Stokes drift that transport plastics. In addition to the indirect influence of the wind (i.e. forcing ocean surface currents and generating waves), the wind can also have a direct influence on the transport of buoyant objects through "windage", where the wind acts on an object's surface area exposed to air (Richardson, 1997). Currents, wind, and waves have large temporal and spatial variations, both

horizontally and vertically. Ocean surface currents typically have a pronounced vertical profile, with current speeds decaying rapidly with depth (Laxague et al., 2017) and which can significantly influence on the transport of buoyant objects (van der Mheen et al, 2020b). The different forcing mechanisms that transport buoyant plastics are rarely independent and interact with each other in complex ways. In addition, depending on their specific characteristics (e.g., size, shape, density), buoyant objects can react differently to the same forcing conditions (Maximenko et al., 2012).

**4.1   Plastic transport and accumulation along ocean fronts**

Plastics entering the ocean from land-based sources are subject to many physical processes on the continental shelf before they being transported into the deeper ocean or beach on coastlines (Figure 3). Several studies investigating plastic debris in the IO have mainly focussed on the biology and biogeochemistry (Roy et al., 2015; Sarma et al., 2015; Sarkar et al., 2018) and only a few studies have addressed the transport of plastics between inshore and offshore regions. Physical processes that lead to

convergent flows, where ocean currents flow towards each other, are one of the most important features for the transport of buoyant plastics. Convergent flows promote downwelling, which causes accumulation of buoyant plastic debris along the convergent flow boundary defined as the front (Figure 3b).

Convergent flows occur along ocean fronts, defined as the boundary between two distinct water masses or a region where the

rate of change of selected physical properties is much greater than the surrounding areas (Bowman and Esaias, 1977; Belkin and Cornillon, 2007; D'Asaro et al., 2018). Fronts occur from hundreds of metres to many thousand kilometres and some are short-lived, but most are quasi-stationary and emerge at the same location on seasonal time scales (Belkin and Cornillon,

2007). Ocean fronts are considered hotspots of marine life (Belkin et al., 2009) because flow convergence at fronts channel nutrients towards the fronts and stimulate increased production at different trophic levels (Owen, 1981; Baltar et al., 2016;

Sarma et al., 2015; Woodson and Litvinb, 2015; Sarkar et al., 2018). Aggregations of plankton, larvae, and eggs are often found on the surface (Figure 3b). Here, as the water sinks at the front due to convergent flow, buoyant material will remain at the surface (Miyao and Isobe, 2016). Predators such as fish and higher order biota are found above and beneath the front. Due to the accumulation of buoyant debris, including plastics, along fronts, the risk of marine organisms interacting with plastics is high.


There are many types of fronts that are formed through different physical processes. They include river plumes (Luketina and Imberger 1988; O'Donnell et al., 1998; Karati et al., 2018; Cole et al., 2020); shelf-sea tidal fronts (Simpson and Hunter, 1974; Nahas et al., 2005; Sharples and Simpson, 2020); shelf break fronts (Sharples and Simpson, 2020); upwelling fronts (Brink, 1987); and fronts formed through interaction between flow and topography (Haury and Hamner, 1986; Pattiaratchi et al.,

1987). Fronts appear as a visible band along the sea surface with differences in temperature and salinity (and as a result, density) on either side of the front. All these different types of fronts occur in the IO and are extremely important for the accumulation of plastics.

Buoyant plastics entering the IO from land-based sources such as rivers may accumulate at many different frontal systems

before being transported offshore (Figure 3). During the Southwest monsoon season (section 4.2), coastal fronts are generally formed by the interaction of two different water masses moving in opposite directions, where the low saline runoff waters from rivers are trapped as a salinity front by the high saline upwelling waters. Examples of such fronts are along the southern coast of Sri Lanka (de Vos et al., 2013). As a result, buoyant plastic debris is trapped in the frontal zone along these water masses (Naidu et al., 2021), referring to locations of future investigations. Naidu et al. (2021) also found that the abundance of plastics,

mainly fibres and fragments, were an order of magnitude higher in the frontal zone compared to outside of it off the west coast of India. Similarly, Hajbane et al. (2021) found that close to an offshore reef located in the eastern IO, concentrations of plastics along a coastal front were up to two orders of magnitude higher than in surrounding waters. Frontal systems can also be eroded under storm conditions (e.g. Chen et al., 2020) and as a result, plastics accumulated along coastal fronts may beach on coastlines.


## 4.2 Indian Ocean surface dynamics, plastic transport pathways, and surface accumulation

Compared to other ocean basins, the IO has several unique topographic, atmospheric, and oceanic features that influence the transport of buoyant plastics. To the north, the IO is bounded by the Indian subcontinent, to the west by Africa and the Middle East, to the east by Indonesia and Australia, and in the south by the Southern Ocean. There are two distinctive tropical basins in the northern IO: the Arabian Sea in the west and the Bay of Bengal in the east. The IO is connected to the Pacific Ocean through the Indonesian Archipelago that allows tropical water inflow from the Pacific Ocean. In the southern IO, the eastern boundary formed by the African continent does not extend beyond 35°S, which allows for a connection between the southern IO and the South Atlantic Ocean (Gordon, 2003; Lutjeharms, 2006) and facilitates plastic transport between these oceans leading to debris exchange between the garbage patches (van der Mheen et al., 2019; section 4.3). Because the subtropics in the northern IO is covered by land mass, there is no subtropical gyre. Instead, temperature differences between the northern land mass and the ocean drive the monsoon system, which dominates the atmospheric and oceanic dynamics in the region. These dynamics determine the transport of buoyant plastics in the IO (van der Mheen et al., 2020a) and are described in more detail here.

### 4.2.1 Northern Indian Ocean surface dynamics, plastic transport pathways, and beaching

The atmospheric dynamics in the IO are characterised by bi-annually reversing monsoon winds due to seasonal differential heating and cooling of the continental land mass and the ocean (Schott et al., 2009). The Southwest (SW) monsoon generally operates between June and October, and the Northeast (NE) monsoon operates between December through April (Tomczak and Godfrey, 2003). The transition periods between the monsoon seasons are the First Inter-Monsoon (May) and Second Inter-Monsoon (November). The influence of the monsoon system is not limited to the northern IO. Unlike in the other oceans, there are no steady equatorial easterly trade winds in the northern IO. Instead, they only have an easterly component during the NE monsoon season and are westerly during the rest of the year. In addition, the SW monsoon season starts with strengthened south-easterly trade winds in the southern IO (Findlater, 1969; Joseph and Sijikumar, 2004).

These strong, seasonally reversing winds drive the ocean surface currents and circulation patterns in the northern IO (Stramma et al., 1996; Shankar and Shetye, 1997; Schott and McCreary, 2001; Shenoi et al., 2004). During the SW monsoon season, the flow in the northern IO is predominantly towards the east (Figure 4a), from the Arabian Sea into the Bay of Bengal, and the westward flowing North Equatorial Current (NEC) does not exist (Schott et al., 2009). Along the coastlines of India and Sri Lanka in the Arabian Sea, the West Indian Coastal Current (WICC) flows southwards along the western Indian coastline and joins the eastward flowing Southwest Monsoon Current (SMC). The SMC flows from the Arabian Sea past Sri Lanka and into the Bay of Bengal (de Vos et al., 2014). After passing the coast of Sri Lanka, the ocean surface currents form an anti-clockwise eddy called the Sri Lanka Dome (SLD; Su et al., 2021). The western arm of this eddy drives a southward current along the eastern coast of Sri Lanka; the remainder flows northwards along the eastern Indian coastline as the East Indian Coastal Current

(EICC). At the eastern boundary, the South Java Current (SJC) is variable but flows predominantly to the northwest along the Java coast (Sprintall et al., 2009), in the same direction as the Indonesian Throughflow (ITF), which is strongest during the SW monsoon (Sprintall et al., 2009). At the western boundary of the northern IO, the Somali Current (SC) flows north-eastwards during this season.

During the NE monsoon season, the ocean surface currents reverse direction and the flow in the northern IO is predominantly towards the west (Figure 4b), from the Bay of Bengal towards the Arabian Sea (Schott et al., 2009). Both the WICC and the EICC along the Indian coasts reverse direction and the SMC reverses and becomes the Northeast Monsoon Current (NMC), which is weaker than the SMC (de Vos et al., 2014). Together with the East African Coastal Current (EACC), the SC supplies the eastwards South Equatorial Counter Current (SECC). The SECC feeds into the SJC, which flows south-eastwards along the coasts of Sumatra and Java during this season. Another unique feature in the equatorial IO is the strong eastward flowing Wyrtki jets (Wyrtki, 1973) that develop along the equator during the First and Second Inter-Monsoon. These jets are at their strongest during the Second Inter-Monsoon (Qui and Yu, 2009).

The main features to note from the description of the ocean surface currents in the northern IO is the reversal of major currents with the monsoon season. Particle tracking simulations illustrate that these reversing currents transport buoyant objects between the eastern and western section of the northern IO with Sri Lanka and the Maldives in the central section (Figure 5; van der Mheen et al., 2020a). In addition, the net surface transport in the northern IO is eastwards (Schott et al., 2009) because of the strong eastwards SMC, and the eastward flowing Wyrtki jets between monsoon seasons. This net eastward transport, combined with eddies developing year-round in the Bay of Bengal and potentially trapping plastics, result in plastics being present in the Bay of Bengal throughout the year. In contrast, the Arabian Sea is mostly depleted of plastics during the SW monsoon season (Figure 5). However, if sources of plastics increase it is likely that concentrations will also increase. As plastics are transported back and forth between the Arabian Sea and the Bay of Bengal, they frequently come close to coastlines. Because of this, it is likely that a large amount of plastic beaches and accumulates on northern IO coastlines (van der Mheen et al., 2020a). Simulations by van der Mheen et al. (2020a) showed that shorelines of countries bordering the Bay of Bengal were most affected by beaching plastics (Figure 6). This is most likely due to the large source locations of particles (Figure 1b) and ocean dynamics of the Bay of Bengal, resulting in plastics being present in the Bay throughout the year.

The dynamics of beaching plastics is complex and strongly influenced by small-scale coastal ocean dynamics (Isobe et al., 2014), as well as local morphology of the coastline (Zhang, 2017). It is unknown what the influence is of these small-scale effects on basin-scale beaching patterns in the IO. Finally, plastics do not necessarily remain beached indefinitely but can also re-float and re-enter the ocean (Zhang, 2017; Lebreton et al., 2019). Several recent studies highlight the potential of oceanic islands to act as transitory repositories for plastic debris (Monteiro et al., 2018; Pham et al., 2020). As a result, it is unknown

how much plastic is stored on coastlines in the IO, as well as how permanent this sink is. Thus, long-term (multi-decadal) monitoring and field studies are necessary along IO coastlines.

### 4.2.2  Southern Indian Ocean surface dynamics and subtropical garbage patch

In the southern IO, similar to other ocean basins, there is a wind-driven subtropical gyre (Figure 4) which in this instance has several unique features. The gyre is bounded in the north by the South Equatorial Current (SEC), which flows westwards and is relatively steady all year round. At the western boundary of the subtropical gyre, the Agulhas Current (AC) flows poleward along the southern African coastline (Beal et al., 2011). The Agulhas Retroflection (AR) and Agulhas Leakage (AL) provide a connection between the southern IO and South Atlantic Ocean (Gordon, 2003; Lutjeharms, 2006). At the eastern boundary, the Leeuwin Current (LC) flows poleward along the western Australian coastline. This is opposite to the equatorward flow expected for a wind-driven subtropical gyre (Pattiaratchi and Woo, 2009). The Flinders Current (FC; Middleton and Cirano, 2002) flows westward from the Pacific Ocean to the southern IO along the southern Australian coastline. In the south, the gyre is bounded by the Antarctic Circumpolar Current (ACC). The South Indian Counter Current (SICC) flows eastward through the centre of the subtropical gyre (Lambert et al., 2016), which is opposite to the direction expected from Sverdrup theory (Palastanga et al., 2007; Wijeratne et al., 2018). Similar subtropical counter-currents exist in the other oceans but the SICC is unique because it flows across the full width of the basin and splits into three separate branches as it flows between the southern tip of Madagascar and the western coast of Australia (Menezes, 2014).

Most plastic waste enters the southern IO from the Indonesian Archipelago (Jambeck et al., 2015; Lebreton et al., 2017; Schmidt et al., 2017; 2018; Meijer et al., 2021; section 2). These plastics are transported westwards across the IO basin by the SEC and into the subtropical southern IO (van der Mheen et al., 2020c). In the subtropics, converging wind-driven Ekman currents lead to downwelling and associated accumulation of buoyant plastics in a subtropical garbage patch. However, because of the unique features of the southern IO subtropical gyre, the dynamics of the garbage patch in the IO are distinctive. The garbage patch is very sensitive to different transport mechanisms (van der Mheen et al., 2019). Under influence of Stokes drift and wind, van der Mheen et al. (2019) showed that the subtropical garbage patch centres towards the west of the IO basin, and simulated concentrations are at least a factor two smaller than in the garbage patches in the other oceans. In contrast, if no Stokes drift or wind is included in the simulation, a large and stable garbage patch forms that span almost the entire IO basin width. In this case, the simulated concentrations are of the same order as concentrations in the North Pacific subtropical garbage patch. It is however unknown which forcing mechanisms predominantly transport plastics. As a result, it is unclear which of these simulated garbage patches best represents the reality in the subtropical southern IO. Sampling in this region are too scarce to provide any conclusions (see section 3). More measurements are required to determine the dynamics and concentration of plastics in the subtropical IO garbage patch.

.

Plastics can potentially also be transported from the subtropical IO garbage patch into the southern Atlantic and Pacific oceans. van der Mheen et al. (2019) and Dobler et al. (2019) showed that plastics transported by Stokes drift and/or wind can move from the IO garbage patch past South Africa into the South Atlantic Ocean. Maes et al. (2018) suggested that there is also a "super convergence pathway" connecting the southern IO to the South Pacific Ocean. Their particle tracking simulation results showed particles being transported eastwards close to the southern Australian coastline. However, these results are potentially

in contradiction to the westwards flowing FC in this region (Middleton and Cirano, 2002; Wijeratne et al., 2018), and so the existence of a super convergence pathway between the southern IO and the South Pacific Ocean along the southern Australian coast still needs further investigation.

Finally, plastics can also beach in the southern IO. Countries along the eastern African coast, Madagascar, Mauritius, Réunion,

the Cocos (Keeling) Islands, and Christmas Island are potentially heavily affected by beaching plastics (Figure 6, van der Mheen et al., 2020a). In the simulations by van der Mheen et al. (2020a), most of these plastics originated from rivers in Indonesia.

### 4.2.3   Transport of plastics between the northern and southern Indian Ocean

Strong ocean surface currents can act as transport barriers for buoyant objects, preventing buoyant objects from crossing these

365 currents (Bower, 1991; Brambilla and Talley, 2006). Due to strong equatorial currents, ocean surface drifters do not tend to cross the equator (Maximenko et al., 2012). Therefore, it has been suggested that buoyant plastics generally remain in the hemisphere where they originally entered the ocean (Lebreton et al., 2012). However, as previously mentioned, the easterly trade winds are not steady in the IO and as a result, the NEC and SECC are not steady either. The surface waters of the IO also appears more connected between hemispheres than the other oceans (Froyland et al., 2014). Therefore, van der Mheen et al.

(2020a) suggested that plastics may not remain in their original hemisphere in the IO. Particle tracking simulation results by van der Mheen et al. (2020a) showed that buoyant plastics could cross from the northern IO into the southern IO as they are transported by the SJC along the Sumatran coastline (see an example of this happening in Figure 5f). This mainly occurred during the Second Inter-Monsoon in their simulations.

van der Mheen et al. (2020c) confirmed that ocean surface drifters mainly cross from the northern IO into the southern IO on the eastern side of the IO basin, along the Sumatran coastline. Drifters mainly crossed during the Second Inter-Monsoon and the NE monsoon season. van der Mheen et al. (2020c) also showed that ocean surface drifters crossing from the southern IO into the northern IO do so along the western side of the basin, predominantly during the SW monsoon season. Drifters cross

the equator along the Somalian coast, likely as they are transported by the SC, which is directed north-eastwards during the SW monsoon.

Based on these results, it seems possible for buoyant plastics to cross the equator in the IO. It is, however unclear how frequently this occurs.

### 4.3   Sinking and ingestion

#### 4.3.1   *Sinking*

Sinking and settling of plastics on the seafloor due to fragmentation and biofouling may be a major sink of plastic debris in the ocean (Koelmans et al., 2017). Based on deep-sea sediment core samples between 500-1000 m depth in the south-west IO, Woodall et al. (2014) estimated that 4 billion fibres per km$^2$ were present in the IO, but did not report on a mass estimate. Ingested plastics by deep-sea fauna in the IO (Taylor et al., 2016) are also evidence that plastics sink to the seafloor. However, no evidence of the total size of this sink currently exists and the understanding of the exact processes of biofouling, fragmentation, and sinking, as well as the timescales on which these occur is limited.

However, the IO is one of the most productive regions in the global oceans due to intense upwelling during the southwest monsoon (Qasim, 1977). This high surface productivity results in a high export flux of organic particles from the euphotic zone to the deep sea (Ittekkot et al., 1996; Guptha et al., 1997). As a result of this high productivity, biofouling of plastic debris may occur rapidly in the IO. As a result, sinking of plastics due to biofouling may be particularly relevant in the IO.

#### 4.3.2   *Ingestion*

Ingestion of plastics can occur at the ocean surface, in the water column, and on the seafloor. Estimates of plastic ingestion by vertebrates (van Franeker, 2011; Davison and Ash, 2011), indicate that the global ingestion of plastics could be on the same order of magnitude as the amount of plastics accumulating in subtropical garbage patches (van Sebille et al., 2015). However, plastic ingestion is generally considered only a temporary and not a permanent sink of marine plastic debris.

Throughout the IO (Figure 2b), multiple studies have sampled ingested plastics in a variety of different fauna: benthic invertebrates (Taylor et al., 2016; Naidu et al., 2018), sessile invertebrates (Thushari et al., 2017), fishes (Ismail et al., 2018; Karthik et al., 2018; Baalkhuyar et al., 2019; Crutchett et al., 2020; McGregory and Strydom, 2020; Sparks et al., 2020), including large sharks (Cliff et al., 2002), seabirds (Cherel et al., 2017; Cartraud et al., 2019), turtles (Hoaru et al., 2014), bivalves (Naidu, 2019), and corals (Saliu et al., 2019; Patti et al., 2020). Recorded ingestion rates varied widely between species, from only approximately 0.4% of large sharks sampled (Cliff et al., 2002) to up to 90% of fish sampled (Sparks et al., 2020).

These sampling studies are both relatively few and relatively recent, so no estimates can be given about the total amount of plastic ingested by marine fauna in the IO, or any trends in plastic ingestion. Cherel et al. (2017) did find that the wandering albatross chicks they investigated at Crozet and Kerguelen Islands had ingested low plastic loads compared to albatross chicks in the North Pacific Ocean. Crutchett et al. (2020) found low plastic ingestion levels in sardines compared to global levels.

They also suggested that sampling plastics in globally common fishes, such as sardines, is a good way to compare and monitor ingestion rates between different locations around the world. In contrast, Hoaru et al. (2014) found that turtles in the south-west IO had a similar occurrence of ingested plastic as in the Mediterranean, south-west Pacific, and Gulf of Mexico but a much higher amount of ingested plastic by number, volume, and weight.

## 5.  Impacts

Plastic debris can harm the marine environment, communities, and economies in many different ways. The review by Law (2017) discusses the different harmful effects of plastic debris, as well as existing evidence for their impact in detail. This section briefly discusses the harmful effects of plastic debris and provides more detail about the potential impacts specific to the IO and IO rim countries. In section 6.4, we also provide an overview of the different policies and initiatives that are emerging in IO rim countries to prevent, mitigate, or clean-up plastic debris in the IO.

### 5.1  Entanglement

Approximately 100 million tonnes of marine fishes have been landed globally each year since 2000, approximately 13% of which originated from the IO (Pauly and Zeller 2016). The IO supports a range of marine ecosystem services including numerous and expansive fishing ventures with 7%, or one million tonnes, of IO landings considered large commercial pelagic fishes, e.g. tuna and billfishes. The commercial pelagic fisheries of the IO alone are worth over US$1300 million annually.

ALDFG becomes "ghost gear", which continues to entangle and injure or catch wildlife, for decades or centuries as it slowly degrades, increasing the pressure on marine wildlife. Entangled and caught fauna of ALDFG starts initially with sea turtles, marine mammals, sharks, and large predators for the first few months, changing to crustaceans when gear fragment into smaller parts (Macfadyen et al., 2009; Wilcox et al., 2015; Stelfox et al., 2016).

ALDFG affects the tourism industry through a multitude of factors, including the removal of iconic marine species and wildlife in general, as well as contributing up to 90% of shoreline debris and degrading of the perceived beauty of an area (Gunn et al., 2010; Lachmann et al., 2017). Ghost gear is also considered to be destructive to both natural habitats, i.e. coral reefs, and manufactured objects, i.e. boats (Laist and Liffman, 2000; Gunn et al., 2010). These devaluations decrease benefits from coral reef tourism (US$ 36 billion) (Spalding et al., 2017) and global exports of aquarium fish (US$ 15 billion) (Raja et al., 2019).

Data from genetic analyses of Olive Ridley turtles entangled in ghost nets in the Maldives showed that the individual turtles originated from populations nesting in India and Sri Lanka (Stelfox et al. 2020b).  This shows that impacts on charismatic marine species that drive tourism can simultaneously impact multiple economies in the IO rim. Whilst regional data are difficult to obtain, 60% of aquarium fish are exported from developing nations (Raja et al., 2019). Thus, the adverse cultural, societal, and sustenance impacts of ghost fishing may disproportionately affect IO rim developing nations (Wong, 2011; Guillotreau et

al., 2012; Lachmann et al., 2017).

Many of the countries bordering the IO are small island developing states (SIDS) where more than 70% of the population lives along the coast and is highly dependent on  marine ecosystem services (Canales et al., 2017). Recent interviews of fishers by Richardson et al. (2021), which included fishermen from Indonesia along the IO rim, showed that the main reasons for gear

loss reported were bad weather and interactions with wild life respectively. Illegal and deliberate gear discard on the other hand was reportedly low. Furthermore, over half of fishermen interviewed across the world reported being "concerned" or "very concerned" about ALDFG, whereby economic losses scored highest (54%) as an issue of concern followed by environmental harm (41%). The reported loss prevention strategies that scored highest were gear maintenance and training crew in gear management, which provide clear avenues for targeted programs to educate and raise awareness around ALDFG

in low income fisheries, such as in many IO rim economies (Richardson et al., 2021). A single ghost net may decrease fishing revenues by US$ 20 thousand annually (Lachmann et al., 2017) and can consume as much as 30% of landings within a single fishery (Laist and Liffman, 2000). Although ALDFG is thought to diminish in catching efficiency over time, catching potential of gillnets were estimated to be 20-30% of original levels for long periods and at least 5% after more than two years of abandonment (Macfadyen et al., 2009; Stelfox et al., 2016). Some catch rates were estimated to be 81 kg/day for tangle nets

over 1.5 years, an average of 64 kg/day for deep water gillnets over a 45 days and 0.6 kg/day for traps left for 3-6 months (Macfadyen et al., 2009).

## 5.2 Ingestion

As described in section 4.3.2, multiple studies in the IO have found ingested plastic in different marine species. However, as described in the review by Law (2017), confirmed cases of ingestion by themselves do not determine the impact of plastic

debris ingestion. Even confirmed cases of individual deaths are not readily translated to impact on species or ecosystem levels. Recently, Roman et al. (2020) determined which types of plastic debris contribute the most to mortality in marine megafauna (cetaceans, pinnipeds, sea turtles, and seabirds). In the IO plastic ingestion has been studied in sea turtles (Hoaru et al., 2014) and seabirds (Cherel et al., 2017; Cartraud et al., 2019).

For turtles, Roman et al. (2020) concluded that film-like plastics, plastic fragments, and hard plastics were most likely to cause death. Although Hoaru et al. (2014) did not find a relationship between the mortality of loggerhead turtles in the south-west

IO and the type of plastic debris they had ingested, they did find a correlation between the length of debris and mortality. They also suggested that loggerhead turtles are have a relatively high tolerance to the ingestion of plastic debris, only 9% of approximately 450 dead turtles were thought to have been killed directly by the ingestion of plastic.


Roman et al. (2020) found that ingestion of hard plastics was most likely to cause death for sea birds. Cherel et al. (2017) sampled the stomach contents of live wandering albatross chicks on the Kerguelen and Crozet Islands in the IO. They found that approximately 50% of chicks had ingested plastic fragments. However, the plastic content was relatively low and Cherel et al. (2017) suggested that this was unlikely to cause any significant deleterious effects. On Réunion and Juan de Nova,

Cartraud et al. (2019) determined the plastic ingestion of deceased petrels and shearwaters. They could find no correlation between the muscular condition of the birds and the amount of plastic ingested. However, all of the birds likely died as a result of injury after being attracted and disoriented by urban light.

## 6. Emerging policies and initiatives on plastic in Indian Ocean rim countries

Of the top 20 countries ranked by mass in estimated mismanaged plastic waste (MMPW), nine are located along the IO rim

(Indonesia, India, Thailand, Malaysia, Bangladesh, South Africa, India, Pakistan, and Myanmar). Plastic waste generated by these countries amounts to ~15% of the worlds' total MMPW (Jambeck et al., 2015), putting pressure on these countries to address the issue of MMPW. Local and national actions have been the primary approach for mitigating plastic pollution (Vince and Hardesty, 2016). The choice of measures at national or local level is left to the IO rim country administrations - in line with the principle of subsidiary. For example, some countries have refundable deposit schemes for bottles (Schuyler et al.,

2018). Targeted deposit schemes can help reduce littering and boost recycling and have already helped several countries achieve high collection rates for beverage containers (Lavee, 2010; Dace et al., 2013; Schuyler et al., 2018,).

Several African countries have introduced measures to address plastic bag pollution. In August 2017, Kenya introduced a total ban on plastic bags, banning all plastic bags' use, manufacture, and importation for commercial and household packaging.

Anyone found in violation is subject to a fine of approximately US$ 20,000-40,000 and/or one to four years imprisonment, making this ban the toughest in the world. Inland countries such as Botswana, introduced a levy on plastic bags in 2010, while Eritrea banned plastic bags in 2005. Rwanda banned plastic bags in 2008 as part of its Vision 2020 plan for sustainability. In 2007, Uganda introduced a ban of lightweight plastic bags which came into effect that year but was never implemented. Tanzania introduced a ban in 2006 and in South Africa, a bag levy was introduced in 2004 although they were never banned

completely.

In India, plastic pollution is being fought at various levels, by state governments, NGOs, and individuals. Initiatives range from banning of plastic to beach clean-ups. Several Indian states have banned or regulated the use of plastic, but India still struggles to manage its huge plastic waste. For microplastics, and their presence in consumer goods, regulations are still under development. The National Green Tribunal in January 2017 asked the Union Government to test leading cosmetics brands for microplastics. The Bureau of Indian Standards (BIS) classified microbeads as "unsafe" for use in cosmetic products and banned microbeads in cosmetics in October 2017, but was only implemented in 2020.

## 7. Case Study: X-Press Pearl nurdle spill off Sri Lanka

A container ship, *X-Press Pearl*, registered in Singapore, was transporting cargo from Jebel Ali (United Arab Emirates) to Colombo (Sri Lanka) via Hamad Port (Qatar) and Hazira (India). The ship departed the port of Hazira on 15 May 2021 and arrived in Colombo on 19 May and was anchored off Colombo Port ~9.5 km offshore in a water depth of 21 m (Figure 7). On 22 May, the sound of an explosion was heard in cargo hold #2 and the ship was then seen to be on fire. By 24 May, the fire had intensified and was spreading toward the aft of the vessel. A louder explosion was heard on 25 May and all personnel were evacuated. The fire burned for 13 days. On 2 June, efforts to move the ship into deeper waters failed, with the aft portion sinking to the seabed. The vessel was transporting 1,486 containers, with a mixed variety of cargo but included 78 tonnes of plastic nurdles (low-density polyethylene pellets) in three containers released into the ocean.

### 7.1 Oceanographic setting

Ocean currents around Sri Lanka during May are such that they flow south along the west coast of India (West Indian coastal current; Figure 4), cross the Gulf of Mannar, and flow south and east along the west and south coasts of Sri Lanka, respectively (see also de Vos et al., 2014; Su et al., 2021). There is a recirculation, the SLD, to the east of the island (Figures 4 and 7). The SLD connects the water from the west coast to the east coast, with currents flowing south along both coasts (Figure 7). During the initial period of the incident, there were strong onshore winds (> 10 ms$^{-1}$) and swell waves (height ~2 m). This was associated with the beginning of the south-west monsoon and the currents were flowing to the south along the west coast of Sri Lanka at the location of the accident. On 22 May, tropical cyclone Yaas formed in the northern Bay of Bengal and propagated to the north and crossing the coast to the north of Visakhapatnam (India). Examination of the water level records along the east coast of India and at Trincomalee (Sri Lanka) indicated the formation of a continental shelf wave (Huyer, 1990) subsequent to the cyclone making landfall (Eliot and Pattiaratchi, 2010) and propagating southward and clockwise around Sri Lanka. As a consequence of the continental shelf wave, sub-tidal water level at Trincomalee started to increase whilst the mean water level at Colombo was decreasing. This resulted in the currents reversing from that due to the monsoon, and flowing east and north along the south and west coasts of Sri Lankan, respectively, after 30 May (Figure 2). The winds decreased to < 5 ms$^{-1}$ after 2 June.

## 7.2 Impact

Nurdles are small plastic pellets used as raw materials for the manufacture of virtually anything that is plastic (e.g., from plastic bags and bottles to automobile parts) and are by definition classified as microplastics as their size is < 5mm. For comparison, the potential discharge of 78 tonnes of nurdles released by *X-Press Pearl* is the second worst release of nurdles from shipping accidents. This includes the 150 tonnes released into Hong Kong harbour in 2012 during a typhoon, and 49 tonnes released into Durban harbour in South Africa in 2017 (Schumann et al., 2019). On 25 May, large quantities of nurdles were washed up on the beaches closest to where the *X-Press Pearl* was anchored (Figure 8). These were transported onshore because of the strong winds and high waves present at that time. Subsequently nurdles were found along the whole of the west of Sri Lanka (https://storymaps.arcgis.com/stories/054f6746857242128298d53d220d76f0).

We used the Regional Ocean Modelling System (ROMS) configured to the oceans around Sri Lanka (de Vos et al., 2014; Su et al., 2021) to provide surface currents together with the Lagrangian particle tracking model Ichthyop (Lett et al., 2008) to simulate the transport of buoyant nurdles along the coast of Sri Lanka (Figure 9). The model results indicated movement of the nurdle plume southwards with the prevailing currents and by 27 May the plume had extended to the south-west corner of the Island (Figure 9b). On 29-31 May, the nurdle plume had detached from the coast and was moving offshore to the south-east (Figures 9c,d). The currents off the south-east of the island was getting stronger and were flowing to the south. By 2 June, this current has further strengthened and has curved around the island flowing to the east (Figure 9e). By 4 June the currents along the west coast has reversed with the nurdle plume closest to the coast moving north (Figure 9f). The nurdles that were detached from the coast on 29 May continued to be transported offshore and subsequently to the east. A budget of the simulated nurdles indicated that on 11 June 32% of the nurdles released would have beached; 28% to be within the model domain (Figure 9) and 40% to have exited the model domain. A feature of the simulations is that the nurdles were contained at distinct boundaries representing the fronts separating cold and warm water as described in section 4.2. Also the reversal of the surface currents on 2 June and transport northwards along the Sri Lanka coastline was associated with the continental shelf wave that was due to tropical cyclone Yaas (see section 7.1). With time, the nurdles that leave the model domain will make landfall along the countries in the northern Indian Ocean (e.g. Indonesia, India, Maldives, Somalia) because of the reversing monsoon currents in the region (see Figure 5) and will be a visible pollutant on the beaches for many decades to come.

560

## 8. Summary

The Indian Ocean covers~20% of the Earth's surface and, compared to other oceans has unique physical characteristics (northern landmass, connection to the Pacific Ocean through Indonesia, and connection to the Atlantic Ocean past South Africa, Monsoon) that strongly influence the circulation patterns and therefore the fate of plastics. This paper examined the sources, transport, sinks and impacts of plastics in the IO. The main sources of plastics are from rivers (terrestrial sources), with the majority input in the northern Indian Ocean, particularly the Bay of Bengal (Figure 10). There are also land-based sources from Indonesia into the southern Indian Ocean. Inputs from Australia are negligible due to the absence of rivers and low population density in Western Australia. In the northern IO, reversing monsoon driven currents transport plastic material between the Bay of Bengal and the Arabian Sea with many plastics making landfall in Somalia, The Maldives, Sri Lanka, east and west coasts of India, Myanmar and western Sumatera (Figure 10). There is no garbage patch located in the northern IO however, in the southern IO there is evidence for a garbage patch but it is not well defined, with very few measurements indicate that it may extend across the whole length of the ocean basin. The main beaching region in the southern IO is the coast of northern Madagascar. There also could be leakage of material into the southern Atlantic Ocean garbage patch past South Africa.

### 8.1  Knowledge gaps

The main gap in knowledge for the IO is the scarcity of data in both the surface and deeper ocean. Only a few data points are available for the whole IO including along coastlines and the deeper ocean (Figure 2). This is a very large constraint, which needs addressing. As demonstrated in this paper, we can simulate the transport of plastics using numerical models but confirmation of these results is of paramount importance. However, the simulations pinpoint regions that require attention in future investigations. For example, it is unclear which type of simulated garbage patches (leaky with low concentrations and on the western side of the IO basin, or stable with high concentrations spanning the entire width of the IO basin) best represents the behaviour of the garbage patch in the subtropical southern IO. For the same reasons, it is also unknown how much plastic is contained in the subtropical IO garbage patch as well as how long it is likely to remain in this garbage patch (both as a result of escaping the IO into the South Atlantic Ocean and as a result of other factors such as sinking and ingestion of plastics).

This paper's main sources of plastics were derived from rivers mainly in the northern Indian Ocean. There is most likely transport of plastics from south-east Asia through the Indonesian Throughflow (section 4.1). However, these sources are currently undocumented and need to be investigated. Individual events such as the *X-Press Pearl* ship incident off Sri Lanka are also responsible for large input of plastics

Considering the importance of increasing plastic pollution, the dynamic pathways and their fate in the marine environment of the Indian Ocean needs further attention. Further, studies on the ingestion of plastic particles by marine biota and their residence time in seafood in the marine environment will also be useful for food quality and the ecosystem's overall health.

595

The understanding of the exact processes of biofouling, fragmentation, and sinking as well as the timescales on which these occur is limited. As far as we are aware, no studies have focussed on this specific to the IO.

600

**Author contribution**: Conceptualization: CP, MvdM; resources: CP; writing—original draft preparation: CP, MvdM; writing— contributions, review and editing by all authors. All authors have read and agreed to the published version of the manuscript.

**Competing interests**: The authors declare that the research was conducted in the absence of any commercial or financial relationships that could be construed as a potential conflict of interest.

**Special issue statement**: This paper is part of the synthesis of the recent results of the 2nd International Indian Ocean Expedition Program (IIOE-2: https://iioe-2.incois.gov.in/).

610

**Acknowledgements:** MvdM was supported by an Australian Government Research Training Program (RTP) Scholarship, the University of Western Australia (UWA) CFH & EA Jenkins Postgraduate Research Scholarship, and a UWA Ad Hoc Scholarship. SH was supported by the UWA University Postgraduate Award and an Australian Government Research Training Program (RTP) Scholarship. BN was supported by the UKRI funded One Ocean Hub NE/S008950/1.

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

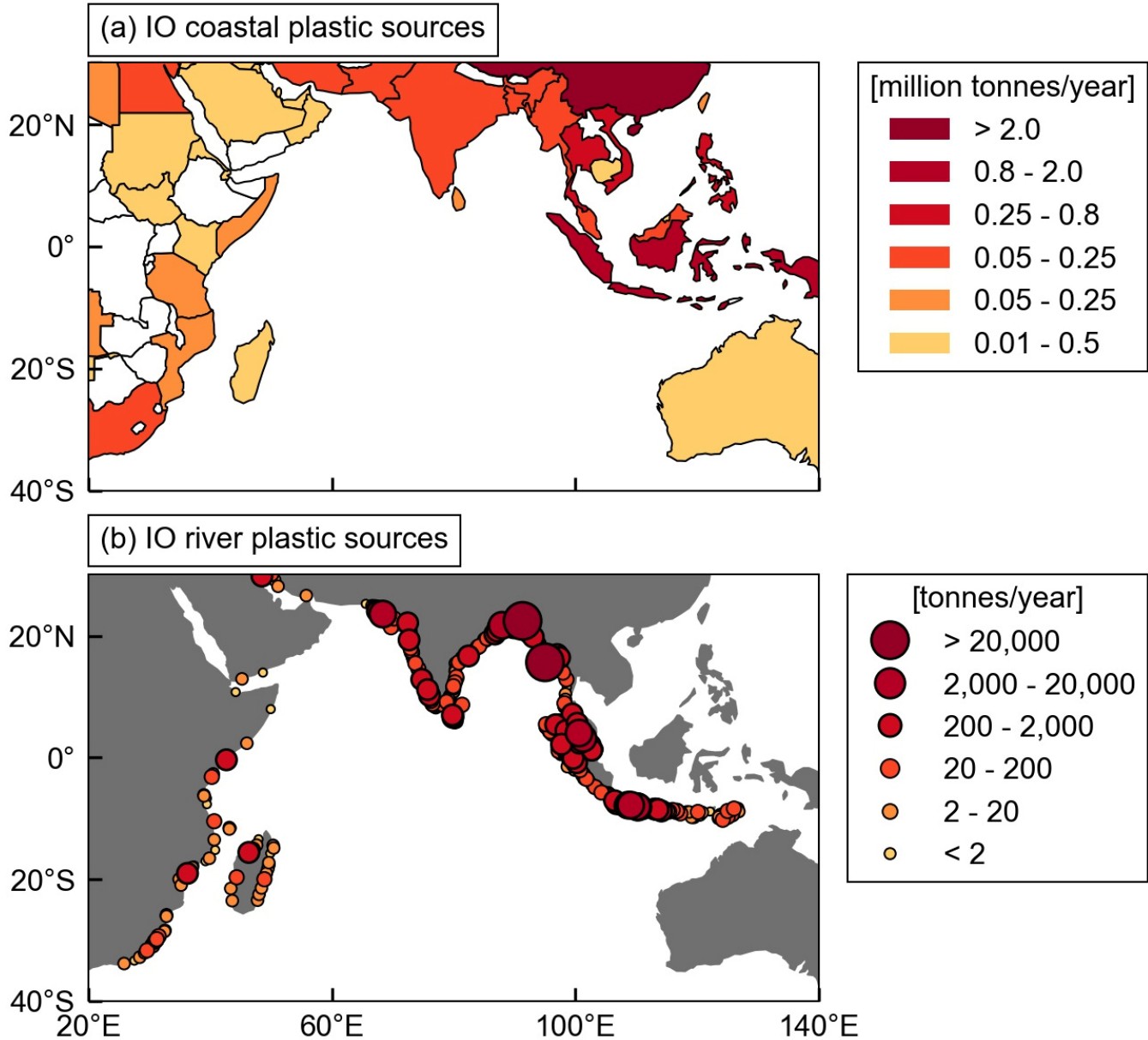

**Figure 1: (a) Estimated coastal sources of plastic waste entering the Indian Ocean, based on data from Jambeck et al. (2015). Plastic waste input estimates for Sri Lanka have been reduced by a factor of 10 in this map to correct for a mistake in Jambeck et al. (2015) data. (b) Estimated river sources of plastic waste entering the Indian Ocean, based on data from Lebreton et al. (2017). The riverine inputs from Australia are negligible due to lack of major rivers. Please note that country boundaries in (a) are from outputs of scientific computer simulations and may not represent political and/or geographical realities.**


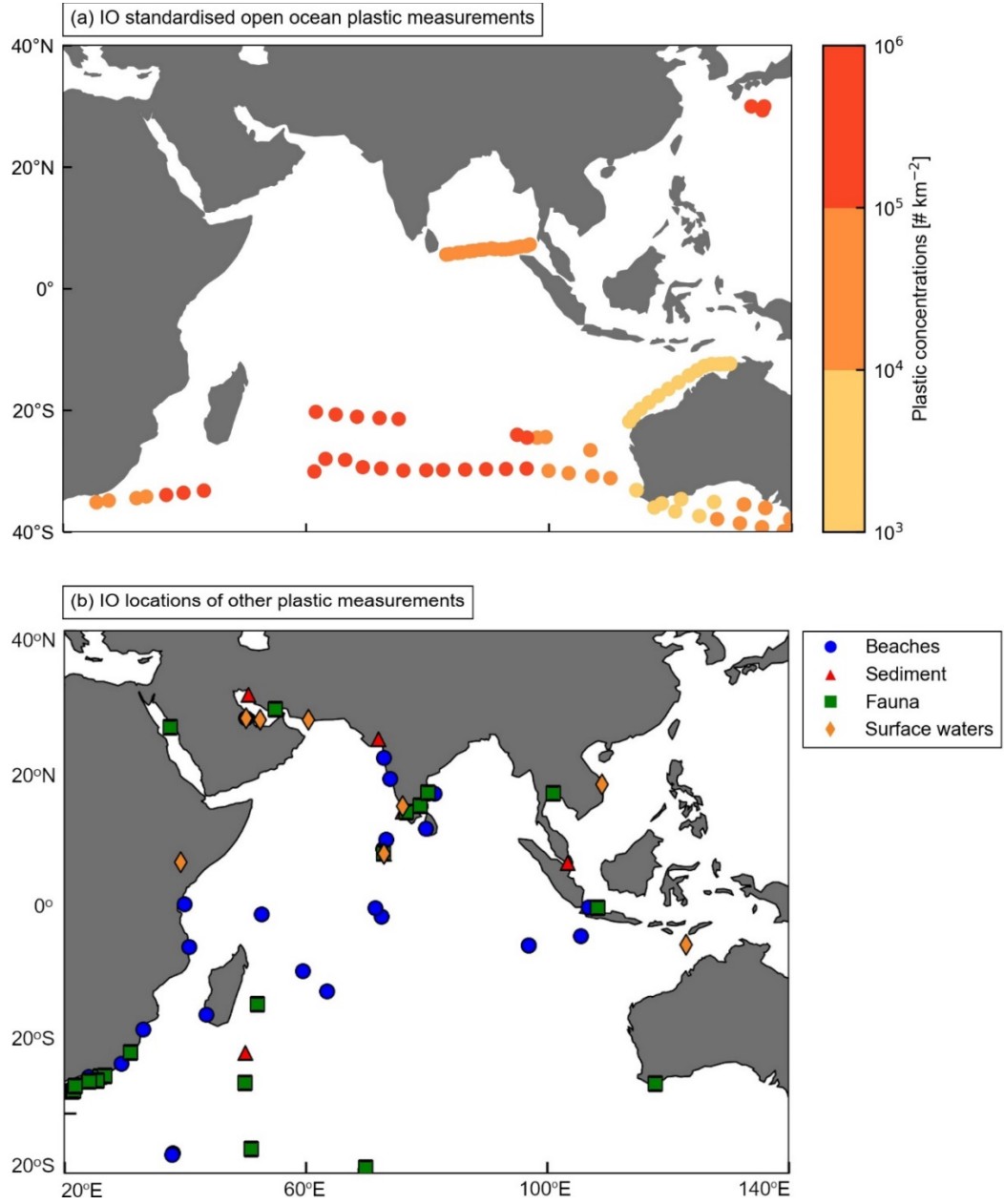

**Figure 2. (a) Standardised measured concentrations of plastics in the Indian Ocean. Original samples were performed by Morris (1980); Reisser et al., (2013); Eriksen et al., (2014); Cózar et al. (2014) and were standardised by van Sebille et al., (2015). (b) Locations of plastic samples taken on beaches, in sediment, and in organisms around the Indian Ocean (references to individual studies listed in Table 1). Because sampling methods vary widely between studies, a quantitative comparison of measured plastic concentrations between these studies is not possible.**

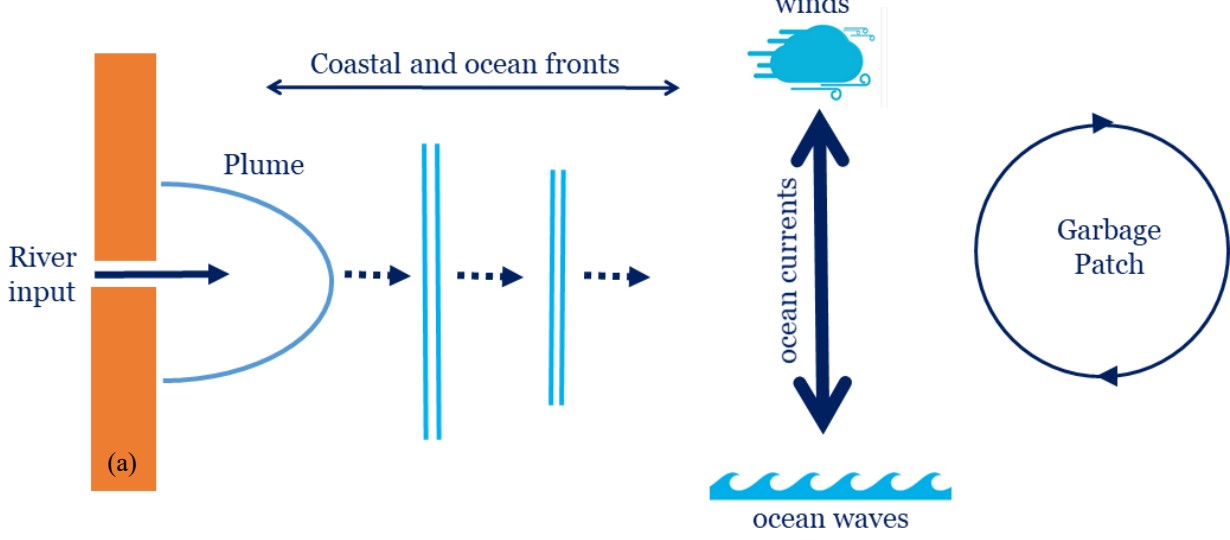

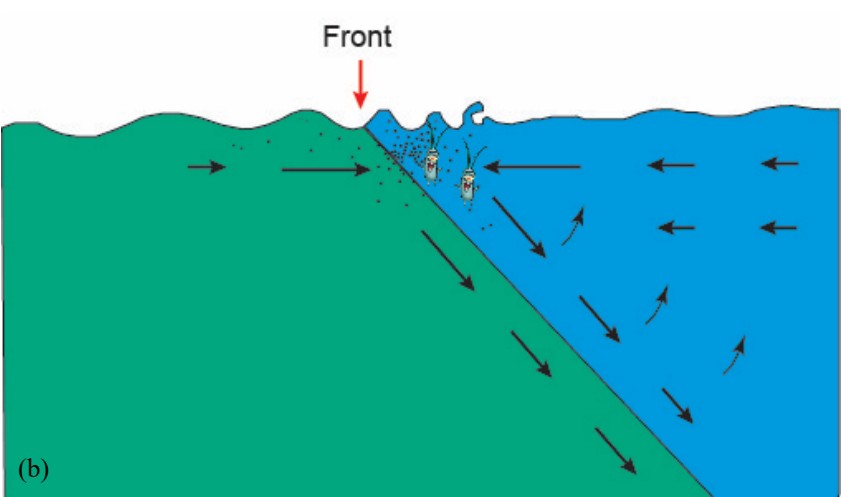


**Figure 3. Schematic illustrating the different physical processes that control the transport of buoyant material from rivers to the open ocean (not to scale): (a) birds eye view; (b) cross-section showing convergence at the front and aggregation of buoyant material. When buoyant material is transported to the open ocean, they are transported by a combination of wind, waves and**

**ocean currents to the centre of ocean gyres forming the garbage patches**

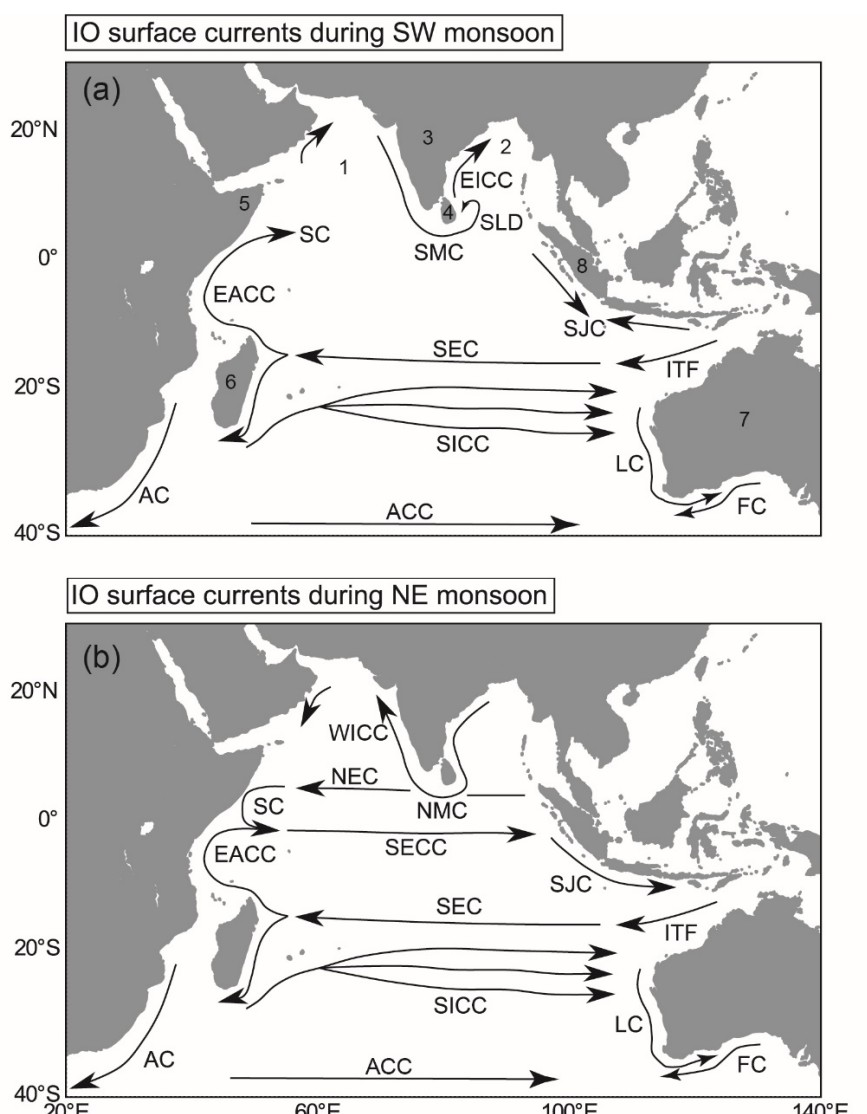

**Figure 4. Schematic of ocean surface currents in the Indian Ocean based on Schott et al. (2009), during (a) the SW monsoon season;**
**and (b) the NE monsoon season. The following currents are shown and labelled with abbreviations: Southwest Monsoon Current**
**(SMC) and Northeast Monsoon Current (NMC); West Indian Coastal Current (WICC) and East Indian Coastal Current (EICC);**
**Sri Lanka Dome (SLD); South Java Current (SJC); Indonesian Throughflow (ITF); Somali Current (SC); East African Coastal**
**Current (EACC); North Equatorial Current (NEC); South Equatorial Counter Current (SECC); South Equatorial Current (SEC);**
**Agulhas Current (AC); Leeuwin Current (LC); Flinders Current (FC); and South Indian Counter Current (SICC). The numbers**
**in (a) refer to marginal seas (1: Arabian Sea; 2: Bay of Bengal) and countries listed in the text: 3: India; 4: Sri Lanka; 5: Somalia;**
**6: Madagascar; 7: Sri Lanka; and, 8: Sumatra (Indonesia).**

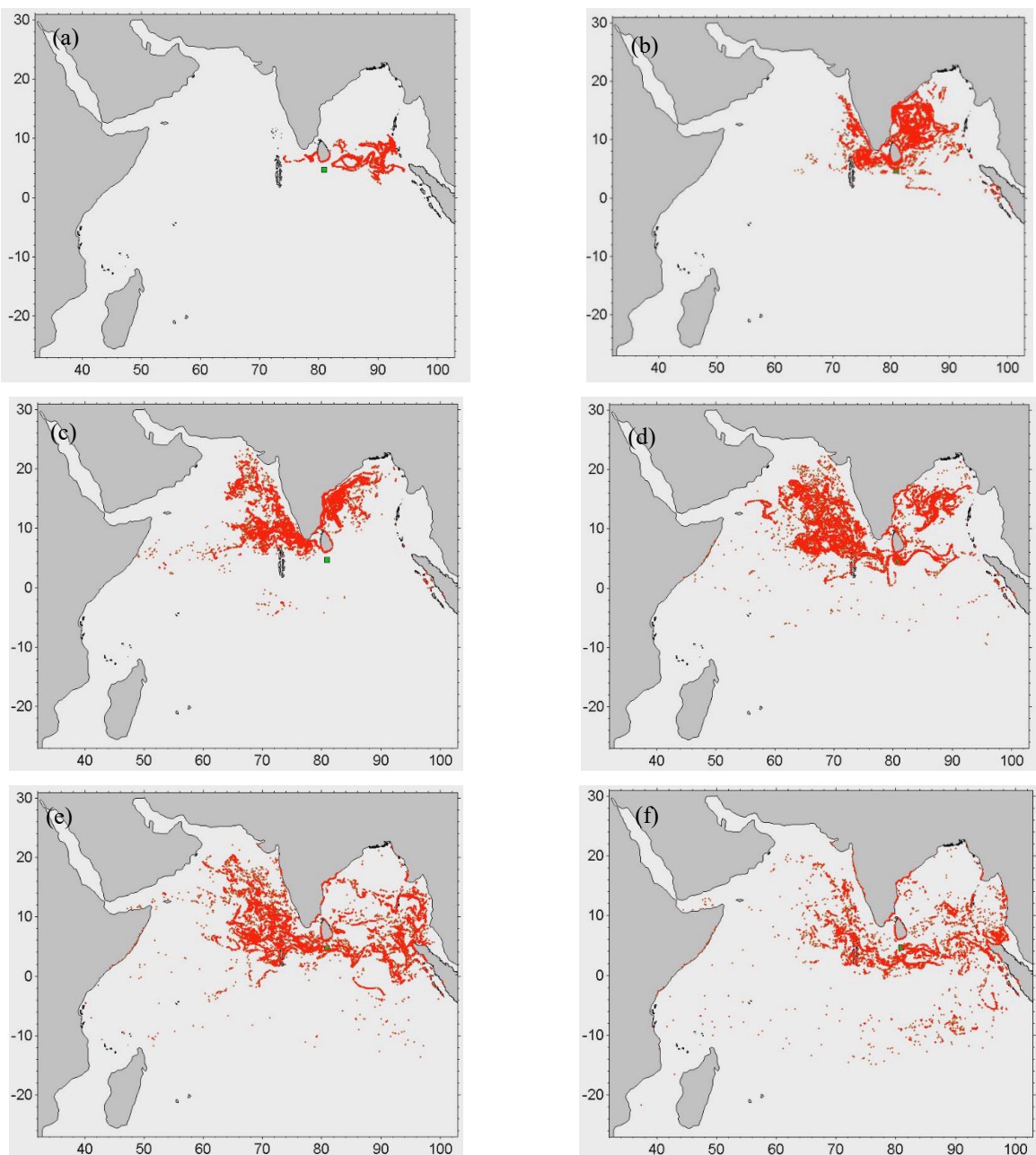

**Figure 5 – Results of a particle tracking simulation using Ichthyop in the northern IO. 10,000 passive particles were released on 1 September to the south of Sri Lanka (green square) and tracked using hourly surface current output from HYCOM model to illustrate the influence of the northern IO surface dynamics on the transport of buoyant plastics.**
**Note that the release location does not represent a location where plastics enter the ocean, but instead was chosen as a central location that is influenced by the reversal of monsoonal currents. (a) 1 November; (b) 1 January; (c) 1 March; (d) 1 May; (e) 1 July; (f) 1 September.**

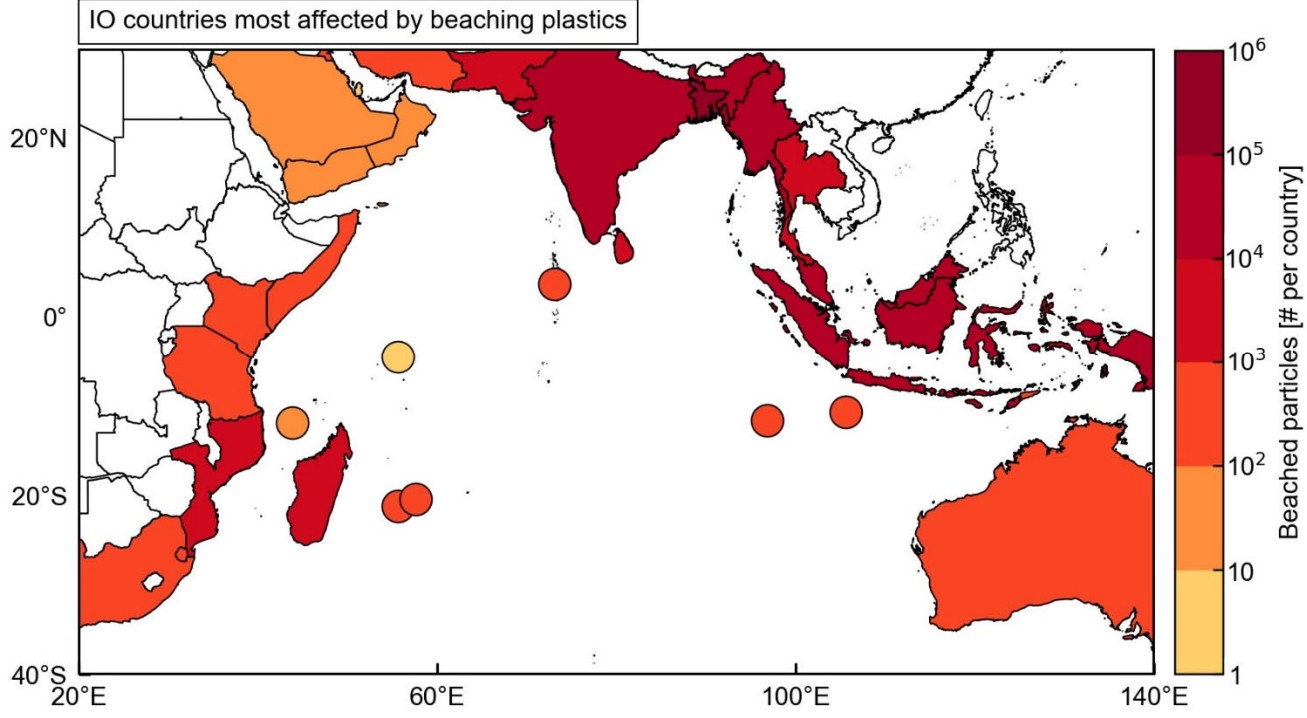


**Figure 6. Countries and islands in the Indian Ocean most affected by beaching plastics from river sources. Colours show the number of particles that have beached in each country in Lagrangian particle tracking simulations by van der Mheen et al. (2020a), where particles beach with a 50% chance if they come within 8 km of a coastline. Figure adapted from van der Mheen et al. (2020a). Please note that country boundaries are obtained from outputs of scientific computer simulations and may not represent political and/or**
**geographical realities.**

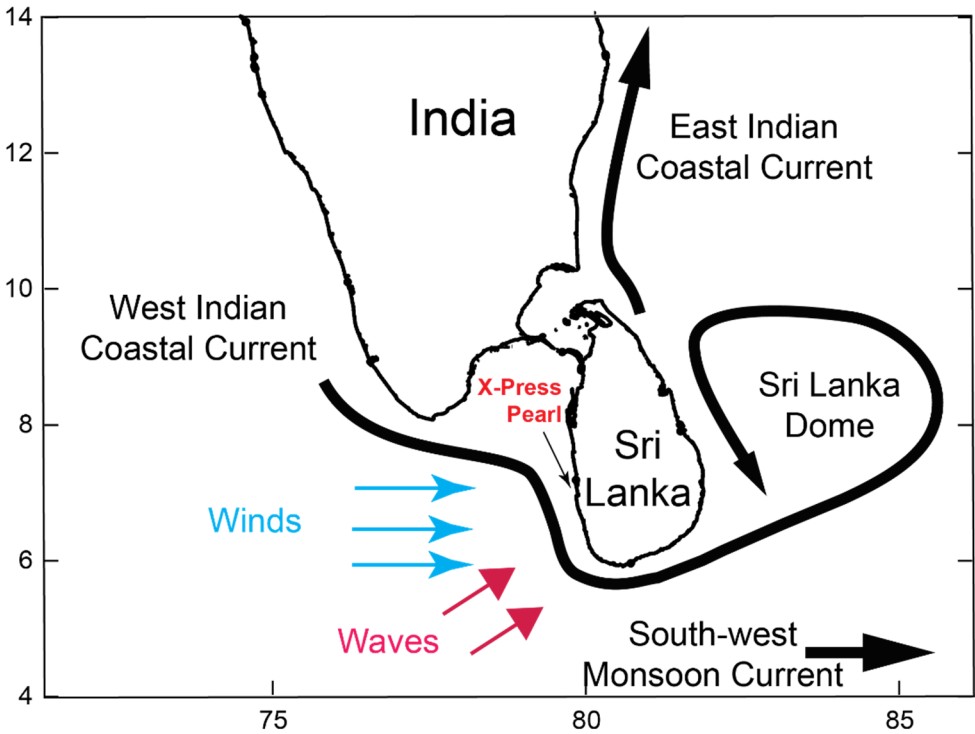

**Figure 7. Location of the *X-Press Pearl* and the major currents in the region. The wind and wave directions experienced during the incident are also shown.**

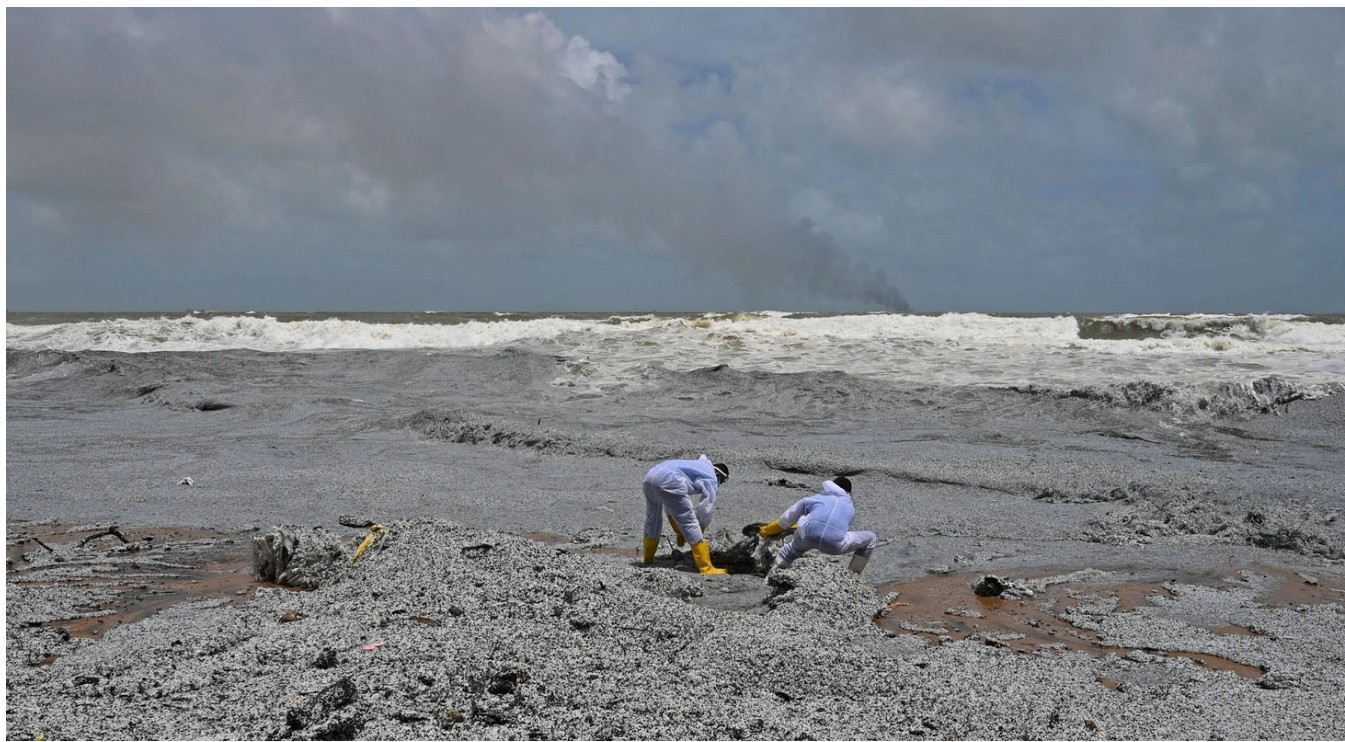

**Figure 8. Image of nurdles on the beach directly to the east of the spill from X-Press Pearl (location in Figure 7).**


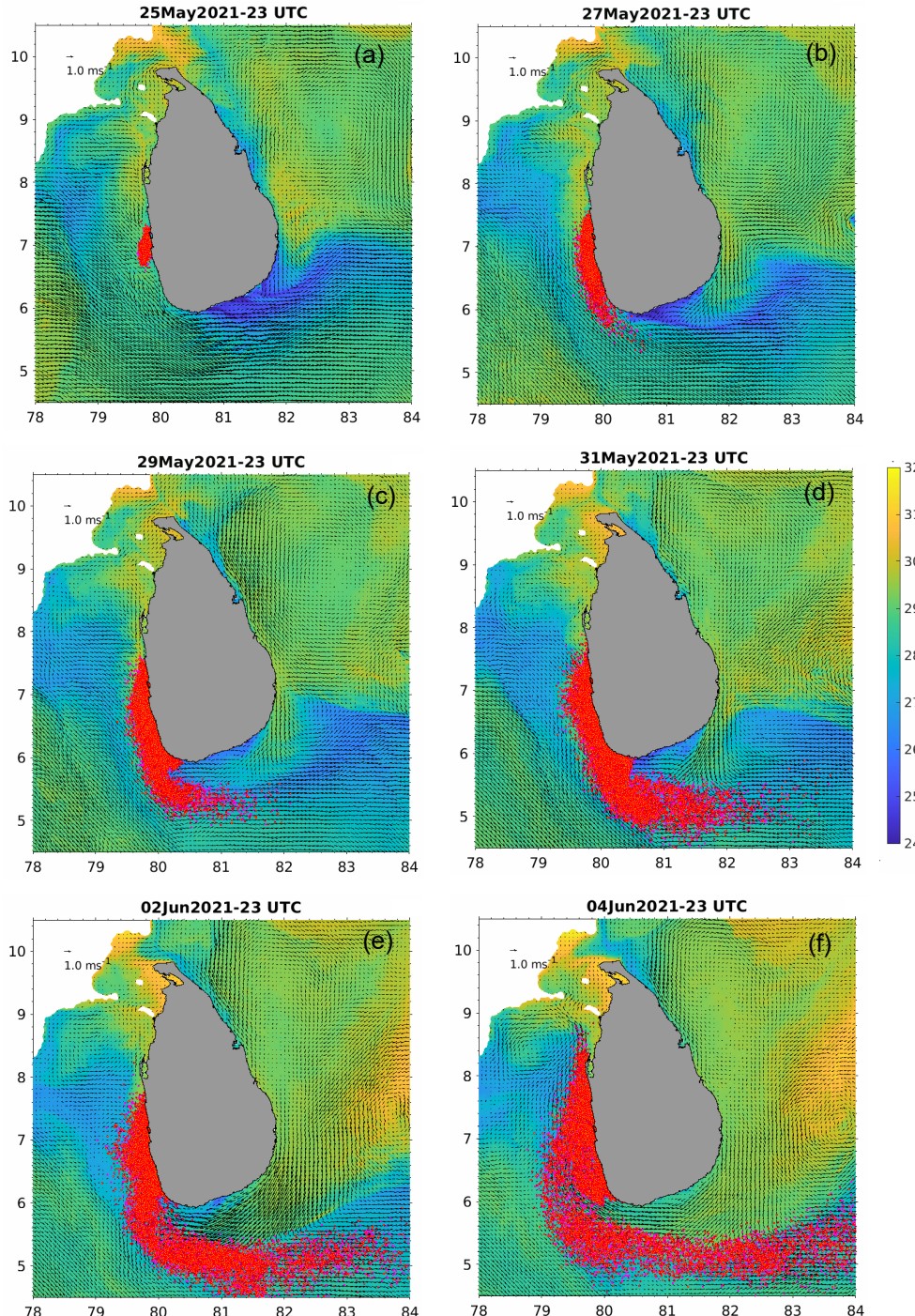

**Figure 9. Time series of the simulated pathway of nurdles from 25 May to 4 June 2021 at 48 hour intervals. The nurdles are shown as red dots, colours are sea surface temperature (⁰ Celsius) and arrows represent surface velocity vectors.**

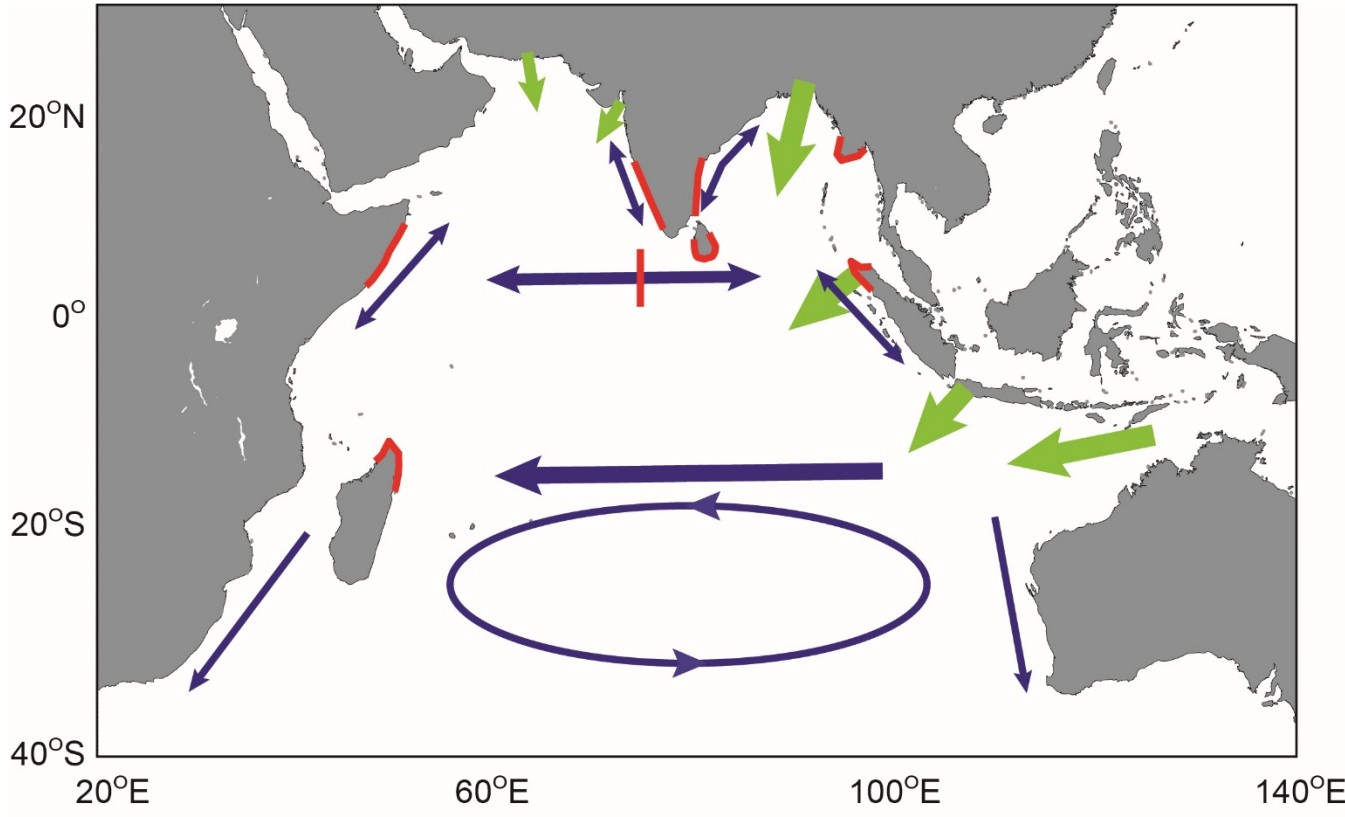

**Figure 10. Schematic showing major plastic sources (green arrows); plastic transport pathways (blue arrows); and major beaching locations of plastics (red regions) in the Indian Ocean.**

**Table 1. Overview of plastic sampling studies performed in the Indian Ocean.**

| Location | Latitude [°N] | Longitude [°E] | Observation site | Categories* | | | Reference |
|---|---|---|---|---|---|---|---|
| | | | | Size [mm] | Shape/ type | Polymers | |
| Surface waters | | | | | | | |
| Northeastern Qatar EEZ | 25.3 | 52.4 | Surface waters | [<0.125, 15.98] | granules, fibres | PP, PE, LDPE, PA, ABS | Castillo et al., 2016 |
| Doha Bay, Qatar | 25.5 | 50.1 | Surface waters | [<0.5, 5] | fibre, film, fragment | LDPE, LDPP | Abayomi et al., 2017 |
| Kuala Nerus and Kuantan, Malaysia | | | Surface waters | < 5 | filaments, fragments, irregular | PES, PE, PA, PVC, PP, PS | Khalik et al., 2018 |
| Faafu Atoll, Maldives | 3.1 | 72.97 | Surface waters | [0.05, 25] | fragments, foils, pellets, fibres, foam | PE, PP, PS, PET, PU, PVC | Saliu et al., 2018 |
| Chabahar Bay, Gulf of Oman (Makran Coasts), Iran | 25.3 | 60.4 | Surface waters | [0.1, 5] | fragment, pellets, fibres paint flakes | PE, PP, PES | Aliabad et al., 2019 |
| Ashmore reef, Australia | -12 | 123 | Surface waters | [0.5, 1.5], [1.5, 5.0], [5.0, 15], >15 | film, fragments, foam | PE, PP | Hajbane et al., 2021 |
| Surface waters and other | | | | | | | |
| Along Cilacap's coast, Indonesia | [-18.4, -7,7] | 109.1 | Surface waters, beaches | [2.5, 5] | - | PP, LDPE, HDPE, PVC, PET, PS, PC | Syakti et al., 2017 |
| Southern coastline of Sri Lanka | - | - | Surface waters, beaches | [1.5, 4.5] | pellets, fragments, films, filaments, foam | PE, PP, PS | Koongolla et al., 2018 |
| Kenyan coastline | 1.7 | 39.2° E | Surface waters, zooplankton | [0.01, 2.4] | fragments, fibres, pellets, film, foam | PP, LDPE; LDPE | Kosore et al., 2018 |
| Maldivian archipelago | 3.1 | 72.97 | Surface waters, corals | < 5 | fragments, films, filaments, foam | PAHs, PE, PP, PA, PS, PU | Saliu et al., 2019 |
| Southwest coast of India | [8.3, 12.8] | [74.9, 77.4] | Surface waters, sediment, fish fauna | [0.3, 5] | pellets, film, fragments, fibres/line, foam | PE, CE, PES, PP, RY | Robin et al., 2020 |
| Beaches | | | | | | | |
| Prince Edward Island | -46.61 | 37.96 | Beaches | >10 | foam, bags, packing | - | Ryan, 1987 |

| Location | Latitude [°N] | Longitude [°E] | Observation site | Categories* | | | Reference |
|---|---|---|---|---|---|---|---|
| | | | | Size [mm] | Shape/ type | Polymers | |
| | | | | | strips, bottles and containers | | |
| Marion Island | -46.84 | 37.85 | Beaches | > 10 | foam, bags, packing strips, bottles and containers | - | Ryan, 1987 |
| Heard Island | -53.00 | 73.48 | Beaches | >10 | foam, bottles, fragments, rope and net fragments | - | Slip and Burton, 1991 |
| Macquarie Island | -54.62 | 158.80 | Beaches | >10 | foam, bottles, fragments, rope and net fragments | - | Slip and Burton, 1991 |
| Transkei, South Africa | -31.75 | 29.38 | Beaches | 1, 1-10, 11-100, 101-1000, >1000 cm$^2$ | plastics, foam, fishing gear | - | Madzena and Lasiak 1997 |
| Jakarta Bay, Indonesia | -5.86 | 106.85 | Beaches | - | plastic bags, footwear, foam, plastic bottles, rope and net fragments | - | Uneputty and Evans 1997 |
| Negombo, Sri Lanka | 7.16 | 79.82 | Beaches | >1 cm$^2$ | - | - | Barnes, 2004 |
| Ari Atoll, Maldives | 3.88 | 72.83 | Beaches | >1 cm$^2$ | - | - | Barnes, 2004 |
| Pemba Island, Tanzania | -5.30 | 39.85 | Beaches | >1 cm$^2$ | - | - | Barnes, 2004 |
| Diego Garcia | -7.34 | 72.51 | Beaches | >1 cm$^2$ | - | - | Barnes, 2004 |
| Christmas Island | -10.54 | 105.59 | Beaches | >1 cm$^2$ | - | - | Barnes, 2004 |
| Cocos (Keeling) Islands | -12.08 | 96.88 | Beaches | >1 cm$^2$ | - | - | Barnes, 2004 |
| Quirimba Island, Mozambique | -12.44 | 40.62 | Beaches | >1 cm$^2$ | - | - | Barnes, 2004 |
| Rodrigues Island | -19.76 | 63.46 | Beaches | >1 cm$^2$ | - | - | Barnes, 2004 |
| Nosy Ve, Madagascar | -23.64 | 43.50 | Beaches | >1 cm$^2$ | - | - | Barnes, 2004 |
| Inhaca Island, Mozambique | -26.02 | 33.00 | Beaches | >1 cm$^2$ | - | - | Barnes, 2004 |
| Mumbai, India | 18.9 | 72.9 | Beaches | [<5, >100] | fragments, pellets | - | Jayasiri et al., 2013 |

| Location | Latitude [°N] | Longitude [°E] | Observation site | Categories* | | | Reference |
|---|---|---|---|---|---|---|---|
| | | | | Size [mm] | Shape/ type | Polymers | |
| The Chagos Archipelago, Chagos (BIOT) | -6 | 71.5 | Beaches | [0.03, 4] | fragments, fibres | NY, PE, PES, PP, RY | Readman et al., 2013 |
| Alphonse Island, Western Indian Ocean, Seychelles | -7 | 52.7 | Beaches | - | footwear, fragments, fishing nets, foam, hard plastic, soft plastic, plastic caps | PET, PVC, PP, HDPE, PE, PS | Duhec et al., 2015 |
| South-east coast South Africa | -34.00 | 24.00 | Beaches | [0.065, 5] | fibres, fragments, foam | - | Nel and Froneman, 2015 |
| Chennai, India | 13.05 | 80.28 | Beaches | - | plastic bags, food wrappers, cups, containers, bottles and caps, rope and net fragments | - | Arun Kumar et al., 2016 |
| St. Brandon's Rock, Mauritius | -16.38 | 59.45 | Beaches | >5 | foam, footwear, plastic, rope | PU, PS, "plastics" | Bouwman et al., 2016 |
| Chennai coast, southeast coast of India | 13 | 81.3 | Beaches | [2, 5] | ovoid, spheroids, disks, cylindrical rods | PE, PP | Veerasingam et al., 2016a |
| Beaches of Goa, India | [15.0, 15.75] | [73.75, 74.25] | Beaches | [1, 5] | cylindrical, spherical, oval | PE, PP | Veerasingam et al., 2016b |
| Coastline of Qatar | 25.5 | 50.1 | Beaches | [<0.5, 5] | fibre, film, fragment | LDPE, LDPP | Abayomi et al., 2017 |
| Vavvaru Island, Maldives | 5.42 | 73.35 | Beaches | [1, 5], >5 | fragments, foils, pellets, fibres, foam | PE, PP, PS | Imhof et al., 2017 |
| Tamil Nadu coast, India | [8,13] | [78, 80] | Beaches | [0.3, 9] | fragments, foam, pellets, film, fibres | PE, PP, PS, NY | Karthik et al., 2018 |
| Cocos (Keeling) Islands | -12.08 | 96.88 | Beaches | [2, 5], >5 | fragments, foam, pellets, bottle caps, plastic bags, packaging, fishing line, | - | Lavers et al., 2019 |

| Location | Latitude [°N] | Longitude [°E] | Observation site | Categories* | | | Reference |
|---|---|---|---|---|---|---|---|
| | | | | Size [mm] | Shape/ type | Polymers | |
| | | | | crates, footwear, straws | | | |
| Coastal areas, Tamil Nadu, India | [8.1, 13.1] | [77.95, 80.5] | Beaches | [0.5, 3] | fibre, fragment, foam, | PE, PP, PS, NY, PES | Sathish et al., 2019 |
| **Sediments** | | | | | | | |
| Alang-Sosiya ship-breaking yard, Gujarath, India | 22 | 72 | Intertidal sediment | [0.0016, 5] | fragments, fibres, films | PU, NY, PES, PS | Reddy et al., 2006 |
| Mangrove sediments, northwest coast of Singapore | 1.2 | | Sediments | [<0.02, 5] | fibres, films, granules | PE, PP, NY, PVC | Nor and Obbard, 2014 |
| Seamounts, Southwest Indian Ocean | -30 | [40, 60] | Sediment | [2, 3] | fibres | RY, ACT, ACR, PES, PA | Woodall et al., 2014 |
| Coastal sediments of Khark Island, Iran | 29.35 | 50.5 | Coastal sediment | [0.1, 5] | fragments, fibres | - | Akhbarizadeh et al., 2016 |
| Vembanad Lake, Kerala, India | [9.5, 10.2] | [76.2, 76.4] | Lake and estuarine sediment | < 5 | fibre, pellet, fragment, foam, film | HDPE, LDPE, PP, PS | Sruthy and Ramasamy, 2016 |
| Faafu Atoll, Maldives | 3.1 | 72.97 | Sediments | [0.05, 25] | fragments, foils, pellets, fibres, foam | PE, PP, PET, PS, PA, PAN, PU, PVC, PVDF | Saliu et al., 2018 |
| Skudai and Tebrau | 1.5 | 103° 24' 85" E | River sediment | [0.1, 5] | fibres/line, fragments, film, foam, beads/ pellets | - | Sarijan et al., 2018 |
| **Fauna** | | | | | | | |
| KwaZulu Natal, South Africa | [-31.2, -29] | [30, 31.5] | Large sharks | - | plastic bags, sheets | - | Cliff et al., 2001 |
| Between Madagascar and Réunion | -22 | 52 | Turtles | >5 | hard plastic, soft plastic, plastic caps, fishing line and rope, foam | - | Hoaru et al., 2014 |
| Southwest Indian Ocean | [-40, -30] | [40, 60] | Benthic invertebrates | - | fibres | PP, PES, ACR, MACR, VI, NF | Taylor et al., 2016 |
| Crozet Islands | -46 | 51 | Seabirds (albatross) | - | fragments | - | Cherel et al., 2017 |

| Location | Latitude [°N] | Longitude [°E] | Observation site | Categories* | | | Reference |
|---|---|---|---|---|---|---|---|
| | | | | Size [mm] | Shape/ type | Polymers | |
| Kerguelen Islands | -49 | 70 | Seabirds (albatross) | - | fragments | - | Cherel et al., 2017 |
| Intertidal area of eastern coastal Thailand | 13 | 101 | Sessile invertebrates | < 0.1 | rod, fragments, fibres | PS, PET, PA | Thushari et al., 2017 |
| Biawak Island, north coast of Indramayu, Indonesia | -5.9 | 108.4 | Below water surface, fish fauna | [1, 5] | fragments, fibres | | Ismail et al., 2018 |
| Tamil Nadu coast, India | [8,13] | [78, 80] | Fish | < 0.5 | fragments, fibres | PE, PP, | Karthik et al., 2018 |
| Réunion | - | - | Seabirds | - | fragments, fibres, films | - | Cartraud et al., 2019 |
| Juan de Nova Island | - | - | Seabirds | - | fragments, fibres, films | - | Cartraud et al., 2019 |
| Frenchman Bay, Western Australia | -35.08 | 118 | Fish (sardines) | <1, <2, >5 | fibres | PP, NY, PE | Crutchett et al., 2020 |
| Algoa Bay, South Africa | -34 | 26 | Fish (South African mullet) | - | fibres, fragments | - | McGregory and Strydom, 2020 |
| Agulhas Bank, South Africa | -36.42 -33.84 -34.58 -36.33 -35.5 -34.78 | 21.33 26.67 25.44 21.49 21.87 24.07 | Fish | <125, [125, 500], [500, 1000], [1000, 2000], >2000 μm | fragments, fibres | - | Sparks and Immelman 2020 |
| Coastal waters of Kochi, Southeastern Arabian Sea, India | 9.9 | 76.6 | Benthic invertebrates | - | particles, fibres | PP | Naidu et al., 2018 |
| Iranian coast of the Persian Gulf, Iran | 26.97 | 55.0 | Molluscans | [0.01, 5] | fragments, fibres, films, pellets | PE, PET, NY | Naji et al., 2018 |
| Coast of Red Sea, Saudi Arabia | [20, 28] | [35, 40] | Fish fauna | [1, 3] | fragments, films, fibres | PP, PE, PS, PVC, PAN | Baalkhuyur et al., 2019 |
| Fishing harbour of Chennai, Southeastern Arabian Sea, India | 13.2 | 80.3 | Bivalves | [0.005, 0.03] | particles, fibres, colorants | PP and colorants** | Naidu, 2019 |
| Maldivian archipelago | 3.1 | 72.97 | Corals | - | - | MEP, DEP,DBP, MEPH,BBz P* | Saliu et al., 2019 |
| **Other** | | | | | | | |

| Location | Latitude [°N] | Longitude [°E] | Observation site | Categories* | | | Reference |
|---|---|---|---|---|---|---|---|
| | | | | Size [mm] | Shape/ type | Polymers | |
| Super markets in Mumbai, India | - | - | Commercial salts | [< 0.021, 7] | fragments, fibres | PES, PET, PA, PE, PS | Seth and Shriwastav, 2018 |
| Asaluyeh city, Iran | 27.53 | 52.6 | Air and street dust | [<0.1, >1] | fragments, fibres, films, spheroids | - | Abbasi et al., 2019 |


*denotes phthalate esters (BBzP), **colorants (dyes or pigments). acrylonitrile butadiene styrene (ABS); modified acrylic (MACR); polycarbonate (PC); acetate (ACT); acrylic (ACR); acrylonitrile butadiene styrene (ABS); benzyl butyl phthalate (BBzP); bis(2-ethylhexyl) phthalate (DEHP); di-butyl phthalate (DBP); di-ethyl phthalate (DEP); di-methyl phthalate (MEP); high density polyethylene (HDPE); low density polyethylene (LDPE); low density polypropylene (LDPP); mono (2-ethylhexyl) phthalate (MEPH); natural fibres (NF); nylon (NY); phthalic acid esters (PAHs); polyacrylonitrile (PAN); polyamide (PA); polyester (PES); polyethylene (PE); polyethylene terephthalate (PET); polypropylene (PP); polystyrene (PS); polyurethane (PU); polyvinylchloride (PVC); rayon (RY); viscose (VI).