# Peer review of "Plastics in the Indian Ocean – sources, transport, distribution and impacts"

_Ocean Science, 2020_

## Referee Comment (RC2)

**General comments**

This paper reviewed the research on marine plastics in the Indian Ocean (IO). Focusing fields include the source, observations, transportation, fate, and impacts of marine plastics. Although the authors should check this manuscript warily because of many mistakes (e.g., not accurate section number, no figure 3), this paper contributes to understanding marine pollution by plastics in IO; hence, I recommended publishing this paper after careful and sincere revisions.

**Specific comments**

| Location | Sentence | Comments / Question / Suggestion |
|---|---|---|
| Abstact | In the northern Indian Ocean, the majority of the plastic material will most likely end up being beached due to the absence of a sub-tropical gyre, | This leads to misunderstanding. Why plastic materials being beached due to the absence of a subtropical gyre. You must explain more for this reasoning. |
| L97-98 | Plastic waste enters the IO from coastal sources transported by wind and tides, from sources far into the hinterland transported by rivers, and directly from ocean-based sources. | Because the authors ignore "the coastal source transported by wind and tide," please explain its meaning in the following subsection. |
| L129 | Lebreton et al. (2017) estimated that plastic waste input from rivers in the IO peaks in August (Figure 1c). | Where is Figure 1c? If the aouthor mean Figure 3 in Lebreton et al. (2017, https://www.nature.com/articles/ncomms15611.pdf), modify the sentence. If not so, show Figure 1c. |
| L 130 | In the southern hemisphere, the largest coastal and riverine sources of 130 IO plastic waste are from Indonesia and eastern Africa (Figure 1b). | I could not understand why the authors mean "the largest coastal and riverine source of IO plastic waste are from Indonesia and eastern Africa." For me, the largest looks like Indonesia only. |
| L 170 | This therefore highlights the need for a standardised global protocol for the study | Already some researchers focus on the standardization of protocols. Refer them, for example: |

| | | |
|---|---|---|
| | of plastic debris and should be a major priority in ocean plastic research going forward. | Michida Y., Chavanich S., Chiba S., Cordova M.R., Cózar Cabañas A., Galgani F. Hagmann P., Hinata H., Isobe A., Kershaw P., Kozlovskii N., Li D., Lusher A.L., Martí E., Mason S.A., Mu J., Saito H., Shim W.J., Syakti A.D., Agung Dhamar, Takada H., Thompson R., Tokai T. Uchida K. Vasilenko K., Wang J (2020) Guidelines for Harmonizing Ocean Surface Microplastic Monitoring Methods. Ministry of the Environment Japan, 71 pp.

Isobe A., Buenaventura N.T., Chastain S., Chavanich S., Cózar A., DeLorenzo M., Hagmann P., Hinata H., Kozlovskii N., Lusher A.L., Martí E., Michida Y., Mu J., Ohno M., Potter G., Ross P.S., Sagawa N., Shim W.J., Song Y.K., Takada H., Tokai T., Torii T., Uchida K., Vassillenko K., Viyakarn V., and Zhang W. (2019) An interlaboratory comparison exercise for the determination of microplastics in standard sample bottles. Mar. Pollut. Bull., 146, pp. 831–837. https://doi.org/10.1016/j.marpolbul.2019.07.033.

Gago J., Filgueiras A., Pedrotti M.L., Suaria G., Tirelli V., Andrade J., Frias J., Nash R., O'Connor I., Lopes C., Caetano M., Raimundo J., Carretero O., Viñas L., Antunes J., Bessa F., Sobral P., Goruppi A., Aliani S., Palazzo L., de Lucia G.A., Camedda A., Muniategui S., Grueiro G., Fernandez V., Gerdts G. (2018) Standardized protocol for monitoring microplastics in seawater. JPI-Oceans BASEMAN project. pp. 34. |
| L188 to L201 | Buoyant plastics drifting  (Maximenko et al., 2012). | this paragraph is redundant. Please organize a little more. |
| L 191 | Ocean surface currents are forced by many different | How waves force ocean currents? I think it is because of storks drift. Why the author divide Coriolis force |

| | mechanisms such as wind, waves, tides, and density gradients (Talley et al., 2011; van Sebille et al., 2020). In combination with the Coriolis force, these forcing mechanisms result in Ekman currents, geostrophic currents, and Stokes drift that transport plastics. | and geostrophic currents? If readers are not physical oceanographers, these two sentences lead to misunderstanding. So, please modify them. |
|---|---|---|
| L 203 | - | Where is Figure 3 |
| L 249 | The presence of the land mass in the northern IO results in there being no subtropical gyre. | This explanation is too direct and incorrect. Refer the comments for the abstract |
| L 301 | This location was selected as a central location where current reversals driven by the monsoon, but it does not reflect a source of plastics (see section 4). | Where is the location in section 4? Now I'm reading section 4. |
| L360 - L380 | Subsection 4.3 To the best of our ~ needs further investigation. | Although I could understand what the author means, the explanation looks de-organized. Please modify. |
| L 400 to L 405 | However, ~ in the IO. | The discussion is too rough. Please explain more details. |
| L 413 to L 440 | 5. Fate | What is the difference from Section 4? Section 4 and Section 5 look similar to each other. Perhaps, reorganization of the section is required to help readers' understanding. |
| L 547 | The main beaching region in the southern IO is the coast of northern Madagascar. | Why can readers understand northern Madagascar has a beach region from sections as mentioned above? |
| Figure 3 | | The authors do not refer to this figure in the manuscript. Refer to this figure to the proper place. |

| | | |
|---|---|---|
| | | In figure 3(a), the left side is the land (river); in contrast, in figure 3(b), the left side implies offshore. Please use the same direction in (a) and (b). |
| | | The meaning of the arrow (ocean currents) in (a) is difficult to understand. |
| Table 1 | A sequence of the location | Why do the authors choose this sequence? Arrangement with Observations (this might be "Observation site"?) is more fruitful for readers. |

*Technical corrections*

| Line | Sentence | Comments / Question / Suggestion |
|---|---|---|
| L152 | Size categories as defined by GESAMP (2018; Frias and Nash, 2019) are: <0.1 mm (nanoplastics); 0.33–1.00mm (small microplastics); 1.01–4.75mm (large microplastics); 4.76–200 mm (mesoplastic); and, > 0.200 mm (macroplastics). | Followings are mistakes. 4.76–200 mm (mesoplastic) > 0.200 mm (macroplastics) I recommend using the latest version of GESAMP. GESAMP(2019) http://www.gesamp.org/publications/guidelines-for-the-monitoring-and-assessment-of-plastic-litter-in-the-ocean |
| L 155 | high- and low density polypropylene (HDPP and LDPP, respectively); | I have no experience using high- and low-density polypropylene. I do not think it is not shared. Check Figure 2.1 in GESAMP (2019). |
| L159 | However, all types of plastics were found in water and sediment samples (fibres, fragments, films, and pellets). | What about Foam? Check Figure 9.4 in GESAMP (2019). |
| L165 | Global open ocean plastic samples were standardised by van Sebille et al. (2020) and the plastic concentrations from these samples in the IO can be | In Figure 2a, the authors refer van Sebille et al. (2015). Which is the right? |

| | | |
|---|---|---|
| | quantitatively compared (Figure 2a). | |
| L 220 | Convergent flows promote downwelling causing an accumulation along the convergent flow boundary of buoyant plastic debris. | I recommend inserting "front" here. |
| L215 | Aggregations of plankton, larvae, and eggs are often found on the surface. Here, as the water sinks at the front due to convergent flow buoyant material will remain at the surface. Predators such as fish and higher order biota are found above and beneath the front. | I recommend referring to the paper to strengthen the importance of fronts.

Miyao Y., and Isobe A. (2016) A combined balloon photography and buoy-tracking experiment for mapping surface currents in coastal waters. J. Atmos. Oceanic Technol., 33, pp. 1237–1250. https://doi: 10.1175/JTECH-D-15-0113.1. (see Fig 5) |
| L253 | 4.2.1 Northern Indian Ocean surface dynamics and plastic transport pathways | The font in the other sections (e.g., 4.2.2) is italic. |
| L 266 | Along the coastlines of India and Sri Lanka in the Arabian Sea, the West Indian Coastal Current (WICC) | No WICC in Figure 4. |
| L269 | After passing the coast of Sri Lanka, the ocean surface currents form an anti-clockwise eddy called the Sri Lanka Dome (SLD; Su et al., 2021). | No SLD in Figure 4 |
| L300 | Passive particles (100,000) were released at a location to the south of Sri Lanka (Figure 4) on 1 Sep 2019 (end of the south-west monsoon) and tracked over a period of 12 months. | The authors used Figure 4; is it a mistake of Figure 5? |
| L 302 to | During the first two months of ~ | Is Figure 4 a misrefer of Figure 5? |

| | | |
|---|---|---|
| L 313 | and Indonesia (Figure 4e). | |
| L 324 | In the south, the gyre is bounded by the Antarctic Circumpolar Current (ACC). | I recommend adding ACC in Figure 4. |
| L 347 | Mheen et al. (2020a) showed that buoyant plastics can cross from the northern IO into the southern IO as they are transported by the SJC along the Sumatran coastline. This mainly occurred during the Second Inter-Monsoon in their simulations. | If need, I recommend referring to Figure 5. |
| L360 | To the best of our knowledge, no studies have currently focussed on the transport of plastics from the Pacific Ocean into the IO through the ITF. | Perhaps, the words are no need to explain. |
| L 372 to L380 | Based on Lagrangian particle tracking simulations, Maes et al. (2018) suggested ~ still needs further investigation. | Do you mean the pathway through FC? If so, use FC elsewhere. |
| L 550 | 7.2 Knowledge gaps | Where is 7.1? |
| L567 | colourants | additivities? |
| Figure 4 | | The authors should add more information (national, currents, date) to figure for easy understanding. |
| Figure 7 | | Brown looks like Red. Change color. |
| Table 1 | Naidu, , 2019 | Naidu, 2019 |
| Table 1 | Barnes,(2004 | Barnes, 2004 |
| Table 1 | Nel and Froneman 2015 | Nel and Froneman, 2015 |

---

## Author Comment (AC1)

**Response to reviewers: (Manuscript ID: os-2020-127)**

**Plastics in the Indian Ocean – sources, transport, distribution and impacts**

We would like to thank and acknowledge the reviewer for their careful reading and constructive comments on the manuscript. We believe that we have addressed the issues raised by the reviewer and the proposed changes to the manuscript are detailed in this document. We trust that the reviewer and the editor will find that the suggested changes will make the manuscript suitable for publication.

Please note that the line numbers referred to in this document are those in the original manuscript commented by the reviewers.

| # | Reviewer comment | Author response |
|---|---|---|
| | **Abstract** | |
| 1 | L33: "Some of the highest plastic-polluted rivers end up in the IO with all this…" – this sentence reads a little awkward to me. Suggest changing to something along the lines of "Some of the most plastic-polluted rivers empty into the IO suggesting the IO…" | We have changed this sentence to: "Some of the most plastic-polluted rivers empty into the IO" as suggested. |
| 2 | L48: slight issue with the phrasing, for example discuss vs discussed and identify vs identified. Pick one and be consistent throughout. | We have replaced all past-tense phrases with present tense phrases in the abstract. |
| | **Introduction** | |
| 3 | L33-46: The first paragraph is fascinating; however, it feels inappropriate for this paper (and journal - sorry) given these statements are focused on a historical account of the evolution of plastics in the late 1850s. Even if this were condensed significantly (which I would argue it needs to be, at 14 lines of text it feels too long and detailed – for example, "billiard balls" are mentioned five times), I'm still not convinced it's the best fit. Instead, could you provide historical context for plastic usage in the SE Asia/IO region? I've not seen this information compiled/synthesised in other papers, so that would be a useful contribution. | We have shortened this section to: "Historically, the motivation for the development of synthetic materials like plastics was for the conservation of elephants that inhabit countries along the Indian Ocean (IO) rim in southern Asia and Africa (Freinkel, 2011). The first plastic materials were advertised as saviours of the environment, because it would no longer be necessary to ransack the environment for scarce natural resources (Meikle, 1997). However, the production of plastic materials has increased exponentially since the 1950s (PlasticsEurope, 2019) and plastics have instead become a ubiquitous environmental pollutant (Law, 2017)." |
| 4 | L49 (and some of the sentences in the paragraph above; also line 57 "35% of all plastic materials"): references are somewhat minimal and/or missing in a few places. For example, this sentence "Since a large percentage of all plastics are single use, "throwaway" packaging items, plastic waste has increased at a similar rate. | We have added in references where relevant. For example: "Since a large percentage of all plastics are single use packaging items (PlasticsEurope, 2019), plastic waste has increased at a similar rate (Geyer et al., 2017)." "Around 35% of all plastic materials produced globally have densities higher |

| | | than that of seawater (PlasticsEurope, 2019)"
"Plastics also accumulate biofouling while in the ocean, which can change the overall density and lead to plastics moving vertically in the water column (Lobelle & Cunliffe, 2011; Long et al., 2015; Kooi et al., 2017)." |
|---|---|---|
| | **Section 2 Sources** | |
| 5 | L97-100 – no reference(s) provided | We have added references as follows: "Plastic waste enters the ocean from coastal sources transported by wind and tides (Jambeck et al., 2015), from sources far into the hinterland transported by rivers (Lebreton et al., 2017; Schmidt et al., 2017; 2018), and directly from ocean-based sources (Richardson et al., 2019)." |
| 6 | L102 – "the total amount of plastic waste produced in 2010 by the USA and China" Here and elsewhere, how do these values such as these compare to more recent (2015) estimates? http://advances.sciencemag.org/content/advances/3/7/e1700782.full.pdf | We have added:
"More recently, Kaza et al. (2018) estimated that the total amount of plastic waste produced by IO rim countries in 2016 was around 24 million tonnes, compared to 34 million tonnes by the USA and 39 million tonnes by China." |
| 7 | L110-114 – suggest providing more information on how and when this error was identified, and more importantly, how it has been rectified (this could potentially be included as Supp Info). The level of detail provided here is a little lacking. For example, what exactly is the error originating from the World Bank Data? | We have changed this part to contain more detailed information and the corrected estimate of plastic waste input from coastal populations in Sri Lanka as follows:
"However, it is likely that the estimated amount of plastic waste entering the ocean by Jambeck et al. (2015) for Sri Lanka is incorrect. Jambeck et al. (2015) based their estimate on a reported 5.1 kg of municipal solid waste generated per person per day in Sri Lanka (Hoornweg and Bhada-Tata, 2012). The updated report by Kaza et al. (2018) and dataset available through the World Bank (What A Waste Global Database) indicates that only 0.34 kg of municipal solid waste is generated per person per day in Sri Lanka; this number is also more in line with the amount of waste generated in other developing countries. Using this correction, the amount of plastic waste entering the ocean from Sri Lanka through coastal populations is estimated between 0.021 and 0.057 million tonnes in 2010, instead of between 0.24 and 0.64 million tonnes as reported in Jambeck et al. (2015)." |

| 8 | L123-124 "The estimates of the amount of plastic waste entering the oceans through rivers by Lebreton et al. (2017) and Schmidt et al. (2017) agree relatively well with each other. In contrast, the estimates by Jambeck et al. (2015) of the amount of plastic waste entering the oceans through coasts are an order of magnitude higher" – it's useful to synthesise these 3 studies like this, but I'm left wondering what the take home message is beyond what you've stated here. Could the authors make some sort of recommendation on how the plastics community should move forward, in light of this? Do we need another of these modelling papers to try and figure out who is "most right" or is the more useful path forward to fill an obvious data gap that would help refine one of the existing models? As the authors are aiming to synthesise information and "recommend future research strategies" it would be useful to answer the "now what" question. | Two new papers have been published with estimates of the amount of plastic waste entering the ocean through rivers since we submitted this manuscript. We have included these two papers in our new manuscript:

"More recently, Meijer et al. (2021) estimated that between 0.80 and 2.7 million tonnes of macroplastics (defined by Meijer et al., 2021 as larger than 5 mm) enter the global ocean per year. In this estimate, Meijer et al. (2021) took into account the spatial variability of mismanaged plastic waste generated within a river basin, as well as more advanced climate and terrain characteristics than considered in the estimates of Lebreton et al. (2017) and Schmidt et al. (2017). They calibrated their estimates based on visual sampling of macroplastics at river mouths around the world."

We have also expanded this section to recommend some future research strategies:
"The estimates of the amount of plastic waste entering the oceans through rivers by Lebreton et al. (2017), Schmidt et al. (2017), and Meijer et al. (2021) agree relatively well with each other. In contrast, the estimates by Jambeck et al. (2015) of the amount of plastic waste entering the oceans through coasts are an order of magnitude higher. In even starker contrast, Weiss et al. (2021) re-evaluated the estimates of Lebreton et al. (2017) and Schmidt et al. (2017) and suggested that only 6.1 thousand tonnes of microplastics (defined by Weiss et al., 2021 as smaller than 5 mm) enter the ocean through rivers each year, which is 2 to 3 orders of magnitude smaller than previous estimates. These differences highlight the extreme uncertainty involved in determining the amount of plastic waste entering the ocean from land-based sources. These estimates are based on few measurements of plastics entering the ocean (in the case of Jambeck et al., 2015, only on data from the San Francisco Bay; in the case of Lebreton et al., 2017; |

| | | Schmidt et al., 2017; Meijer et al., 2021; and Weiss et al., 2021, on 30 to 340 samples from 13 to 89 rivers around the world). None of these samples of were taken in IO rim countries or in rivers that empty into the IO. Expanding on these datasets will likely help improve these estimates, especially for the IO. However, as Weiss et al. (2021) demonstrate, to reduce extreme errors it is essential to use comparable sampling methodologies and to collect not only data on the amount of plastics sampled but also on their weight. Furthermore, Meijer et al. (2021) emphasize the importance of sampling plastics at river mouths to get a more reliable estimate of the amount of plastic that actually enters the ocean. However, sampling plastics further upstream in addition to the river mouth, can help improve models of the probability for plastic to reach the ocean from inland areas." |
|---|---|---|
| 9 | L135 – I find the wording of this sentence to be in an odd order, sorry "Although the International Convention prohibited the dumping of waste from vessels in 1988 for the Prevention of Pollution from Ships (MARPOL)" | We have rephrased this sentence as: "In 1988, the International Convention for the Prevention of Pollution from Ships (MARPOL) prohibited the dumping of waste from vessels. However, accidental losses and illegal dumping still contribute to plastic debris." |
| 10 | L136 – what about ghost nets in Carpentaria, do you have any information on whether some could make their way across to the IO? https://www.sciencedirect.com/science/article/pii/S0016718516302603 | Gulf of Carpentaria is a sink. During the monsoon period (Austral summer) the winds are from the south-west to westerly along the north-west coast of Australia and Indonesia – thus the net movement is into Gulf of Carpentaria. However, when the winds reverse to south-easterly they do not have much impact on the transport of debris out of Gulf of Carpentaria. |
| 11 | L149 – "Commonly used type categories are plastic fibres, fragments, films, and pellets" this is a brief and somewhat uninformative statement as it lacks references and other information. For example, why are these the commonly used categories (why does this matter to the reader)? Consider this paper, or others like it: Serra-Gonçalves, C., Lavers, J.L., Bond, A.L., 2019. Global review of beach debris monitoring and future recommendations. Environ. Sci. Technol. 53, 12158-12167. | We have rewritten this section and now refer to relevant existing review papers instead: "Samples of plastic debris consist of different plastic polymers and are generally classified into different type and size categories. Size and type categories can vary widely between authors but it is beyond the scope of this review to discuss these different categories. Instead, we refer to recent review papers by Gigault et al. (2018) and Frias & Nash (2019) discussing plastic size categories, and |

| | | Hartmann et al. (2019) discussing different categories of polymers, sizes, shapes, colours, and origins." |
|---|---|---|
| | | We refer to the suggested paper in the paragraph following it; see our response to review comment #27 for this. |
| 12 | L151 – "Size categories as defined by…" this is actually a very complex and actively debated issue that is often over-simplified. GESAMP may have been (one of) the original groups to define these categories, but there's been much development and learning in the 12 years since the report was published. For example, see Gigault et al. 2018. Current opinion: What is a nanoplastic? Environ. Pollut. 235, 1030-1034. | We have rewritten this section so that it does not contain any specific size categories, but refer to existing review papers discussing this matter in detail instead (see our response to comment #11 for the changes we have made in the manuscript to address this). |
| 13 | L207 – definition of Convergent flows isn't provided until line 210, after it's first mention. This is a little confusing for readers not familiar with this concept | We have included a brief description of what convergent flows are: "Physical processes that lead to convergent flows, where ocean currents flow towards each other, are one of the most important features for the transport of buoyant plastics." |
| | **Section 4** | |
| 14 | **Sections 4.1 to 4.3** – these are well-written and referenced sections. The level of detail is high, explanations are clear, and I found this useful and enjoyable to read. However, it stands out against other sections which, in comparison, are brief and sometimes feel incomplete (or a little unnecessary). I'm not suggesting you write more elsewhere as your article is already 18 pages – instead, is it possible to focus the paper more on these sections where the author's clearly have a wealth of knowledge and experience? (and less on the tangential topics, many of which have already been covered in other papers). | Thank you. We have shortened this section a bit. We have also added the relevant information from section 5 here and removed section 5 (fate), see also our response to comment #15. |
| | **Section 5 (fate)** | |
| 15 | This 1st paragraph is redundant with earlier sections which also talk about buoyant plastics (e.g., line 181-186) and sinks (e.g., line 59, 80-84, and 174). L391-395 – an example of one of the brief sections that seems "thrown in" at the last minute (sorry). While this is interesting and does indeed occur, you either need to provide more information on the mechanism of how this actually occurs, or disregard this entirely and focus on other fates. Two refs that you may want to consider: Cartraud etal. 2019. Plastic ingestion in seabirds of the western Indian Ocean. Mar. Pollut. Bull. 140, 308-314. Fujieda etal. 2008. Ingestion case of plastics by black marlin and lancetfish caught in the east Indian Ocean. Memoirs of Faculty of Fisheries 57, 47-48. | We have removed section 5 (fate) because, as you say, there was a lot of duplicate or irrelevant (for the purpose of this paper) information. We have added the relevant information from this section to section 4. As these changes are quite extensive, we will not list them all here. Instead, please see the manuscript with tracked changes. |
| 16 | Section 5.1 – well-written, however I'm not entirely convinced this section adds anything new as it essentially | Agreed, we have removed this section and kept only a few summary sentences in |

| | summarises the findings of one paper written by the authors (van der Mheen et al. 2019). | section 4. See also our response to comment #15. |
|---|---|---|
| 17 | L490 - Abandoned, lost, and discarded fishing gear (ALDFG) already defined on line 136 | Thank you, we have replaced this with the abbreviation ALDFG. |
| | **Section 6** | |
| 18 | section 6.1 (ghost nests) is 2 paragraphs, but only one sentence (line 489) contains information or direction refers to the Indian Ocean – can you replace some of this with information more specific to the region? | We have added some more information relevant to the IO in this section: "Data from genetic analyses of Olive Ridley turtles entangled in ghost nets in the Maldives showed that the individual turtles originated from populations nesting in India and Sri Lanka (Stelfox et al. 2020b). This shows that impacts on charismatic marine species that drive tourism can impact multiple economies in the IO rim simultaneously." "Recent interviews of fishers by Richardson et al. (2021), which included fishermen from Indonesia along the IO rim, showed that the main reasons for gear loss reported were bad weather and interactions with wild life respectively. Illegal and deliberate gear discard on the other hand was reportedly low. Furthermore, over half of fishermen interviewed across the world reported being "concerned" or "very concerned" about ALDFG, whereby economic losses scored highest (54%) as an issue of concern followed by environmental harm (41%). The reported loss prevention strategies that scored highest were gear maintenance and training crew in gear management, which provide clear avenues for targeted programs to educate and raise awareness around ALDFG in low income fisheries, such as in many IO rim economies (Richardson et al., 2021)." |
| | **Other** | |
| 19 | Acknowledgements - Australia nPostgraduate Award | Thank you, we have corrected this. We have also corrected a few other mistakes in the Acknowledgements section. |
| 20 | **Figure 7** – the brown arrows and red boundaries are a little difficult to distinguish (perhaps even more so when the image is reduced in size during printing). Can you select another colour, being mindful of folks with colour blindness https://www.ascb.org/science-news/how-to-make-scientific-figures-accessible-to-readers-with-color-blindness/ | We have changed the brown arrow to green. |

| 21 | **Table 1** – some entries seem incomplete, plastic size and type data is available at least for Cocos, yes? | We have corrected this and filled in all available information in Table 1. We have also reorganised Table 1, in our response to a comment by another reviewer. |

---

## Author Comment (AC2)

**Response to reviewers: (Manuscript ID: os-2020-127)**

**Plastics in the Indian Ocean – sources, transport, distribution and impacts**

We would like to thank and acknowledge the reviewer for their careful reading and constructive comments on the manuscript. We believe that we have addressed the issues raised by the reviewer and the proposed changes to the manuscript are detailed in this document. We trust that the reviewer and the editor will find that the suggested changes will make the manuscript suitable for publication.

Please note that the line numbers referred to in this document are those in the original manuscript commented by the reviewers.

| # | Reviewer comment | | | Author response |
|---|---|---|---|---|
| 22 | This paper reviewed the research on marine plastics in the Indian Ocean (IO). Focusing fields include the source, observations, transportation, fate, and impacts of marine plastics. Although the authors should check this manuscript warily because of many mistakes (e.g., not accurate section number, no figure 3), this paper contributes to understanding marine pollution by plastics in IO; hence, I recommended publishing this paper after careful and sincere revisions. | | | Thank you. We have done our best to address all mistakes. |
| | **Specific comments** | | | |
| | Location | Sentence | Comments / Question / Suggestion | |
| 23 | Abstract | In the northern Indian Ocean, the majority of the plastic material will most likely end up being beached due to the absence of a sub-tropical gyre, | This leads to misunderstanding. Why plastic materials being beached due to the absence of a subtropical gyre. You must explain more for this reasoning. | Buoyant plastics tend to accumulate in garbage patches in the subtropical gyres. In the northern IO, there is no subtropical gyre because the subtropics is blocked by land. Because there is no subtropical gyre, there is no associated subtropical garbage patch. So, instead of accumulating in a garbage patch, most plastics in the northern IO are likely to end up on land instead.

We have clarified this in the abstract by rephrasing this sentence as: "In the southern IO, plastics accumulate in a garbage patch in the subtropical gyre. However, this garbage patch is not well defined and plastics may leak into the southern Atlantic or Pacific Ocean. In the northern IO, there is no subtropical gyre and associated garbage because the subtropics is blocked by land. Instead, the majority of plastics most likely end up on coastlines." |

| 24 | L97-98 | Plastic waste enters the IO from coastal sources transported by wind and tides, from sources far into the hinterland transported by rivers, and directly from ocean-based sources. | Because the authors ignore "the coastal source transported by wind and tide," please explain its meaning in the following subsection. | The "coastal sources transported by wind and tides" refers to sources from coastal populations (so not transported by rivers, but entering the ocean 'directly' from coastal populations). The plastic waste input into the ocean from these sources were estimated by Jambeck et al. (2015). We discuss this in some detail in the paragraph following this one (under the sub-heading 2.1 Land-based sources). We have highlighted this in the text by adding: "around 15% of global ocean plastic entered the IO directly through coastal sources (Figure 1a)" |
|---|---|---|---|---|
| 25 | L129 | Lebreton et al. (2017) estimated that plastic waste input from rivers in the IO peaks in August (Figure 1c). | Where is Figure 1c? If the author mean Figure 3 in Lebreton et al.(2017, https://www.nature.com/articles/ncomms15611.pdf), modify the sentence. If not so, show Figure 1c. | Thank you for pointing this out; this was an old reference that we did not update correctly. We have removed the reference to Figure 1c from the manuscript. |
| 26 | L 130 | In the southern hemisphere,the largest coastal and riverine sources of IO plastic waste are from Indonesia and eastern Africa (Figure 1b). | I could not understand why the authors mean "the largest coastal and riverine source of IO plastic waste are from Indonesia and eastern Africa." For me, the largest looks like Indonesia only. | We have changed this sentence to: "In the southern hemisphere, the largest coastal and riverine sources of IO plastic waste are from Indonesia (Figure 1)." |
| 27 | L 170 | This therefore highlights the need for a standardised global protocol for the study of plastic debris and should be a major priority in ocean plastic research going forward. | Already some researchers focus on the standardization of protocols. Refer them, for example:

Michida Y., Chavanich S., Chiba S., Cordova M.R., Cózar Cabañas A., Galgani F. Hagmann P., Hinata H., Isobe A., Kershaw P., Kozlovskii N., Li D., Lusher A.L., Martí E., Mason S.A., Mu J., Saito H., Shim W.J., Syakti A.D., Agung Dhamar, Takada H.,Thompson R., Tokai T. Uchida K. Vasilenko K., Wang J (2020) Guidelines for Harmonizing Ocean Surface Microplastic Monitoring Methods. Ministry of the Environment Japan, 71 pp.

Isobe A., Buenaventura N.T., Chastain S., ChavanichS., Cózar A., DeLorenzo M., Hagmann P., Hinata | We have rewritten this section and changed it to: "In contrast, the methods used in the sampling of plastics on beaches and in sediment vary widely (as illustrated in Table 1) and offer only a qualitative confirmation that plastics have been found on beaches and in sediment throughout the IO (Figure 2b). As discussed extensively in the review by Serra-Gonçalves et al. (2019), adopting a standardised framework to collect and report on beach debris is essential for these studies to be of use to the wider scientific community. Isobe et al. (2019) discuss the importance of a standardised protocol for laboratory analysis of plastics." |

| | | | | |
|---|---|---|---|---|
| | | | H.,Kozlovskii N., Lusher A.L., Martí E., Michida Y., MuJ., Ohno M., Potter G., Ross P.S., Sagawa N., Shim W.J., Song Y.K., Takada H., Tokai T., Torii T.,Uchida K., Vassillenko K., Viyakarn V., and Zhang W. (2019) An interlaboratory comparison exercise for the determination of microplastics in standardsample bottles. Mar. Pollut. Bull., 146, pp. 831–837. https://doi.org/10.1016/j.marpolbul.2019.07.033.

Gago J., Filgueiras A., Pedrotti M.L., Suaria G., Tirelli V., Andrade J., Frias J., Nash R., O'Connor I., Lopes C., Caetano M., Raimundo J., Carretero O., Viñas L., Antunes J., Bessa F., Sobral P., Goruppi A., Aliani S., Palazzo L., de Lucia G.A., Camedda A., Muniategui S., Grueiro G., Fernandez V., Gerdts G. (2018) Standardized protocol for monitoring microplastics in seawater. JPI-Oceans BASEMAN project. pp. 34. | We also refer to several review papers that discuss the standardization of plastic size classes as well as different types, etc. Please see our response to comment #11 for this. |
| 28 | L188 to L201 | Buoyant plastics drifting (Maximenko et al., 2012). | this paragraph is redundant. Please organize a littlemore. | We have kept this paragraph in the manuscript, as it is the first time that we address this information. However, we had a lot of redundancy in section 5. We have removed section 5 (fate) completely from the manuscript, see our response to comment #15. |
| 29 | L 191 | Ocean surface currents are forced by many different mechanisms such as wind, waves, tides, and density gradients (Talley et al., 2011; van Sebille et al., 2020). In combination with the Coriolis force, these forcing mechanisms result in Ekman currents, geostrophic | How waves force ocean currents? I think it is because of storks drift. Why the author divide Coriolis force and geostrophic currents? If readers are not physical oceanographers, these two sentences lead to misunderstanding. So, please modify them. | Yes, waves create Stokes drift. Regarding "dividing" Coriolis force and geostrophic currents, we think there is a bit of a misunderstanding here. These two sentences say that wind, waves, tides, and density gradients *together* with the Coriolis force create Ekman currents, geostrophic currents, etc.

This is only meant as a brief summary of the relevant forces to take into account when considering the transport of buoyant plastics. For a more detailed description, we refer to the paper by van Sebille et al. (2020) |

| | | | | |
|---|---|---|---|---|
| | | currents, and Stokes drift that transport plastics. | | as well as others papers. Readers who are not physical oceanographers can refer to these papers if they would like to understand more.

We think this brief summary and the reference to other review papers is sufficient, so we have not made any changes to address this issue in the manuscript. |
| 30 | L 203 | - | Where is Figure 3 | Figure 3 is present in the manuscript, but it was not referred to in the text. We have corrected this, see our response to comment #37. |
| 31 | L 249 | The presence of the land mass in the northern IO results in there being no subtropical gyre. | This explanation is too direct and incorrect. Refer the comments for the abstract | For clarification, we have replaced this sentence with: "Because the subtropics in the northern IO is covered by land mass, there is no subtropical gyre."

See also our response to comment #23. |
| 32 | L 301 | This location was selected as a central location where current reversals driven by the monsoon, but it does not reflect a source of plastics (see section 4). | Where is the location in section 4? Now I'm reading section 4. | This was meant to be section 2. We have corrected this. |
| 33 | L360 - L380 | Subsection 4.3 To the best of our ~ needs further investigation. | Although I could understand what the author means, the explanation looks de-organized. Please modify. | We have removed subsection 4.3 and instead moved only the most relevant information from this subsection to subsection 4.2. As these changes are quite extensive, we will not list them all here. Instead, please see the manuscript with tracked changes. |
| 34 | L 400 to L 405 | However, ~ in the IO. | The discussion is too rough. Please explain more details. | We have changed this section to:

"Sinking and settling of plastics on the seafloor due to fragmentation and biofouling may be a major sink of plastic debris in the ocean (Koelmans et al., 2017). Based on deep-sea sediment core samples between 500-1000 m depth in the south-west IO, Woodall et al. (2014) estimated that 4 billion fibres per km2 were present in the IO, but did not report on a mass estimate. Ingested plastics by deep-sea fauna in the IO (Taylor et al., |

| | | | | 2016) are also evidence that plastics sink to the seafloor. However, no evidence of the total size of this sink currently exists and the understanding of the exact processes of biofouling, fragmentation, and sinking, as well as the timescales on which these occur is limited.

However, the IO is one of the most productive regions in the global oceans due to intense upwelling during the southwest monsoon (Qasim, 1977). This high surface productivity results in a high export flux of organic particles from the euphotic zone to the deep sea (Ittekkot et al., 1996; Guptha et al., 1997). As a result of this high productivity, it is possible that biofouling of plastic debris may occur rapidly in the IO. As a result, sinking of plastics due to biofouling may be particularly relevant in the IO." |
|---|---|---|---|---|
| 35 | L 413 to L 440 | 5. Fate | What is the difference from Section 4? Section 4 and Section 5 look similar to each other. Perhaps, reorganization of the section is required to help readers' understanding. | We have removed section 5 (fate) because, as you say, there was a lot of duplicate information. We have added the relevant information from this section to section 4. As these changes are quite extensive, we will not list them all here. Instead, please see the manuscript with tracked changes. |
| 36 | L 547 | The main beaching region in the southern IO is the coast of northern Madagascar. | Why can readers understand northern Madagascar has a beach region from sections as mentioned above? | We have referred to Madagascar in Figure 4. This from the model results (Figure 7). |
| 37 | Figure 3 | | The authors do not refer to this figure in the manuscript. Refer to this figure to the proper place.
In figure 3(a), the left side is the land (river); in contrast, in figure 3(b), the left side implies offshore. Please use the same direction in (a) and (b).

The meaning of the arrow (ocean currents) in (a) is difficult to understand. | We have now referred to Figure 3 in relevant places in the manuscript.

We have changed the colours in Figure 3b, so that in both sub-figures the left side represents land.

We have added an explanation to the figure caption. |
| 38 | Table 1 | A sequence of the location | Why do the authors choose this sequence? Arrangement with Observations (this might be | We have reorganised Table 1 so that it is sorted by Observation site first and then by publication date. |

| | | | "Observation site"?) is more fruitful for readers. | |
|---|---|---|---|---|
| | **Technical corrections** | | | |
| 39 | L152 | Size categories as defined by GESAMP (2018; Frias and Nash, 2019) are: <0.1 mm (nanoplastics); 0.33–1.00mm (small microplastics); 1.01–4.75mm (large microplastics);4.76–200 mm (mesoplastic); and, > 0.200 mm (macroplastics). | Followings are mistakes. 4.76–200 mm (mesoplastic) > 0.200 mm (macroplastics)

I recommend using the latest version of GESAMP

GESAMP(2019) http://www.gesamp.org/publications /guidelines-for-the-monitoring-and-assessment-of-plastic- litter-in-the-ocean | We have removed this sentence from the revised manuscript; see our response to comment #11. |
| 40 | L 155 | high- and low density polypropylene (HDPP and LDPP, respectively); | I have no experience using high- and low- density polypropylene. I do not think it is not shared. Check Figure 2.1 in GESAMP (2019). | We have removed this sentence from the revised manuscript; see our response to comment #11. The distinction between HDPE/LDPE is still made in a few studies summarized in Table 1. However, we have only made this distinction when the authors themselves do this as well (for papers that do not make this distinction, we have only listed PE as the plastic type in Table 1).

Figure 2.1 in GESAMP (2019) shows PE in the piechart, but in the caption is does mention that this consists of both HDPE and LDPE. So, it seems that making this distinction is the choice of the authors. |
| 41 | L159 | However, all types of plasticswere found in water and sediment samples (fibres, fragments, films, and pellets). | What about Foam? Check Figure 9.4 in GESAMP (2019). | We have removed this sentence from the revised manuscript; see our response to comment #11. We do mention foam under the shape/type column in Table 1, if studies reported this as a separate type. |
| 42 | L165 | Global open ocean plastic samples were standardised by van Sebille et al. (2020) and the plastic concentrations from these samples in the IO can be quantitatively compared (Figure 2a). | In Figure 2a, the authors refer van Sebille et al.(2015). Which is the right? | Thank you for pointing this out. The correct reference is van Sebille et al. (2015), we have corrected this in the revised manuscript. |

| 43 | L 220 | Convergent flows promotedownwelling causing an accumulation along the convergent flow boundary of buoyant plastic debris. | I recommend inserting "front" here. | Agreed and inserted: "accumulation of buoyant plastic debris along the convergent flow boundary defined as the front" |
|---|---|---|---|---|
| 44 | L215 | Aggregations of plankton, larvae,and eggs are often found on the surface. Here, as thewater sinks at the front due to convergent flow buoyant materialwill remain at the surface. Predators such as fish and higherorder biota are found aboveand beneath the front. | I recommend referring to the paper to strengthen the importance of fronts. Miyao Y., and Isobe A. (2016) A combined balloon photography and buoy-tracking experiment for mapping surface currents in coastal waters. J. Atmos. Oceanic Technol., 33, pp. 1237–1250. https://doi: 10.1175/JTECH-D- 15-0113.1. (see Fig 5) | Agreed and inserted: "Here, as the water sinks at the front due to convergent flow, buoyant material will remain at the surface (Miyao and Isobe, 2016)." |
| 45 | L253 | 4.2.1 Northern Indian Oceansurface dynamics and plastic transport pathways | The font in the other sections (e.g., 4.2.2) isitalic. | Thank you, we have changed the font to be italic here too. |
| 46 | L 266 | Along the coastlines of India andSri Lanka in the Arabian Sea, theWest Indian Coastal Current (WICC) | No WICC in Figure 4. | The WICC is shown in Figure 4b. It is not present in Figure 4a because it becomes the EICC during the SW monsoon season. |
| 47 | L269 | After passing the coast of Sri Lanka, the ocean surface currentsform an anti-clockwise eddy called the Sri Lanka Dome (SLD; Su et al., 2021). | No SLD in Figure 4 | The SLD is shown in Figure 4a, it is not shown in Figure 4b because it does not form during the NE monsoon season. There is typo in Figure 4 though, the SLD is referred to as the SD instead. We have corrected this in the figure. |
| 48 | L300 | Passive particles (100,000) were released at a location to the southof Sri Lanka (Figure 4) on 1 Sep 2019 (end of the south-west monsoon) and tracked over a period of 12 months. | The authors used Figure 4; is it a mistake of Figure 5? | Yes, this should be Figure 5. We have removed this paragraph in the new version of the manuscript though (see our response to comment #62). |

| | | | | |
|---|---|---|---|---|
| **49** | L 302 to L 313 | During the first two months of ~ and Indonesia (Figure 4e). | Is Figure 4 a misrefer of Figure 5? | Yes, this should be Figure 5. We have removed this paragraph in the new version of the manuscript though (see our response to comment #62). |
| **50** | L 324 | In the south, the gyre is boundedby the Antarctic Circumpolar Current (ACC). | I recommend adding ACC in Figure 4. | We have included the ACC in Figure 4. |
| **51** | L 347 | Mheen et al. (2020a) showed thatbuoyant plastics can cross from the northern IO into the southern IO as they are transported by the SJC along the Sumatran coastline. This mainly occurred during the Second Inter-Monsoon in their simulations. | If need, I recommend referring to Figure 5. | We have added: "Mheen et al. (2020a) showed that buoyant plastics can cross from the northern IO into the southern IO as they are transported by the SJC along the Sumatran coastline (see an example of this happening in Figure 5f)." |
| **52** | L360 | To the best of our knowledge, no studies have currently focussed on the transport of plastics from the Pacific Ocean into the IO through the ITF. | Perhaps, the words are no need to explain. | We have removed this sentence. |
| **53** | L 372 to L380 | Based on Lagrangian particle tracking simulations, Maes et al.(2018) suggested ~ still needs further investigation. | Do you mean the pathway through FC? If so, use FC elsewhere. | We have removed this section from the manuscript, see our response to comment #33. We have added a shorter description of this pathway to section 4.2.2, it now reads: "Maes et al. (2018) suggested that there is also a "super convergence pathway" connecting the southern IO to the South Pacific Ocean. Their particle tracking simulation results showed particles being transported eastwards close to the southern Australian coastline. However, these results are potentially in contradiction to the westwards flowing FC in this region (Middleton and Cirano, 2002; Wijeratne et al., 2018), and so the existence of a super convergence pathway between the southern IO and the South Pacific Ocean along the |

| | | | | southern Australian coast still needs further investigation." |
|---|---|---|---|---|
| **54** | L 550 | 7.2 Knowledge gaps | Where is 7.1? | We have corrected this. |
| **55** | L567 | colourants | additivities? | corrected |
| **56** | Figure 4 | | The authors should add more information(national, currents, date) to figure for easy understanding. | We have included numbers in Figure 4a and reference these in the caption: "The numbers in (a) refer to marginal seas (1: Arabian Sea; 2: Bay of Bengal) and countries listed in the text: 3: India; 4: Sri Lanka; 5: Somalia; 6: Madagascar; 7: Sri Lanka; and, 8: Sumatra (Indonesia)." |
| **57** | Figure 7 | | Brown looks like Red. Change color. | Colour has been changed |
| **58** | Table 1 | Naidu, , 2019 | Naidu, 2019 | Corrected, thank you. |
| **59** | Table 1 | Barnes,(2004 | Barnes, 2004 | Corrected, thank you. |
| **60** | Table 1 | Nel and Froneman 2015 | Nel and Froneman, 2015 | Corrected, thank you. |

---

## Author Comment (AC3)

**Response to reviewers: (Manuscript ID: os-2020-127)**

**Plastics in the Indian Ocean – sources, transport, distribution and impacts**

We would like to thank and acknowledge the reviewer for their careful reading and constructive comments on the manuscript. We believe that we have addressed the issues raised by the reviewer and the proposed changes to the manuscript are detailed in this document. We trust that the reviewer and the editor will find that the suggested changes will make the manuscript suitable for publication.

Please note that the line numbers referred to in this document are those in the original manuscript commented by the reviewers.

| # | Reviewer comment | Author response |
|---|---|---|
| 61 | The manuscript entitled "Plastics in the Indian Ocean – sources, fate, distribution and impacts" written by Charitha Pattiaratchi et al. is a review of plastic pollution in the Indian Ocean. In general, the manuscript has an excellent proposal to show for the scientific community an actual scenario of plastic pollution in the Indian Ocean, mainly when it has scarce information related to other oceans. The manuscript was organized in the following topics: sources (section 2), observations (section 3), transport (section 4), fate (section 5), impact (section 6), prevention and mitigation (section 6) of plastic debris in the Indian Ocean as well as highlight some of the emerging policies and initiatives, knowledge gaps and recommend future research strategies (section 7) (lines 92-95).

However, the manuscript does not have a section for methodology. Then, it does not possible to know how the authors found the papers for this study.

The authors should be clear in:
what platform of science (Scopus, Scholar Google, Web of Science, Science Direct, and other) these papers were downloaded;
what keywords were used to find the articles;
in what period (time limit) they were downloaded (perhaps from 1980 to 2020 - lines 145-147/Table 01);
What criteria were used for inclusion or exclusion of papers?

These questions must be answered because a review article should provide a comprehensive foundation on a topic, explain the current state of knowledge, identify gaps in existing studies for potential future research, and/or highlight the main methodologies research techniques. The authors tried to do it during the manuscript, but I do not have not access their methodology so I can not able to understand the database of the article to build this study. | We did not include a methodology section as it is a review paper. It is expected that a reader will refer to papers cited for the methodology.

Including a methodology of the different search platforms and search terms as suggested is relevant for a meta-analysis paper but not necessarily for a review paper. |
| 62 | I reinforce again that in a systematic review with a focused question, the research methods must be clearly described. | Agreed, we have removed the description of this simulation |

| | | |
|---|---|---|
| | Besides, a review article does not have an input of new data/results. Therefore, the illustration made by the authors should be excluded from the manuscript (lines 298-315). | from the manuscript. We have, however, kept Figure 5 in the manuscript, as it does not show any new data/results, but serves purely as an illustration of the effect of surface dynamics in the northern IO on buoyant objects. |
| 63 | Also, I think the name of the program is wrong. The correct would not be ichthyop instead of ICHYTOPOP? (line 300). | Yes this should be Ichthyop, thank you for pointing this out. We have corrected this. (Note that we have removed this paragraph from the manuscript, see our response to comment #62, but we do still mention Ichthyop in the caption of Figure 5.) |
| 64 | **2.1 Land-based sources** This topic needs to have an increment of articles, reports, or data from NGOs local, regional about the situation of waste management or plastic pollution in the land. On Scholar Google, I searched these references:
- Vidanaarachchi, C. K., Yuen, S. T., & Pilapitiya, S. (2006). Municipal solid waste management in the Southern Province of Sri Lanka: Problems, issues, and challenges. Waste Management, 26(8), 920-930.
- Talyan, V., Dahiya, R. P., & Sreekrishnan, T. R. (2008). State of municipal solid waste management in Delhi, the capital of India. Waste Management, 28(7), 1276-1287.
- Patti, T. B., Fobert, E. K., Reeves, S. E., & da Silva, K. B. (2020). Spatial distribution of microplastics around an inhabited coral island in the Maldives, Indian Ocean. Science of The Total Environment, 748, 141263. | Thank you for this comment. What we wanted to cover is the inputs of plastics into the ocean rather than issues that deal with the municipal solid waste management issues on land. We feel that this is outside the scope of this review which has an emphasis on the ocean transport. In contrast we have included the third reference in our paper. |
| 65 | **2.2 Ocean-based sources** Oceanic islands act as a source and/or a sink of plastic waste. Different studies in both the Atlantic and the Pacific Ocean have been discussing it and I think it should be discussed or at least presented on this topic. Oceanic Islands could be a temporary reservoir when plastics items fragmenting on beaches, for example, and physical forcing takes out them to water surrounding. On the other hand, plastic items could stay there for a long time on the supratidal zone fragmenting itself (final reservoir) infinitely. I looked for some articles in the Indian Ocean, but I can find nothing. Therefore, I suggest looking for some articles that could bring this discussion.

 On Scholar Google, I searched these references
- Pham, C. K., Pereira, J. M., Frias, J. P., Ríos, N., Carriço, R., Juliano, M., & Rodríguez, Y. (2020). Beaches of the Azores archipelago as transitory repositories for small | Oceanic islands are considered as a land-based source: they are taken into account in for example Jambeck et al. (2015), where data is available for these islands. So we have not included a discussion on oceanic islands specifically in this section.

 However, we agree that they are important (temporary) sinks of plastic. We already discuss this a bit in a later section (now in section 4.2.1 after removal of section 5), we have expanded on this and |

| | | |
|---|---|---|
| | plastic fragments floating in the North-East Atlantic. Environmental Pollution, 263, 114494.
- Monteiro, R. C., do Sul, J. A. I., & Costa, M. F. (2018). Plastic pollution in islands of the Atlantic Ocean. Environmental Pollution, 238, 103-110. | included the two suggested references:
"Finally, plastics do not necessarily remain beached indefinitely, but can also re-float and re-enter the ocean (Zhang, 2017; Lebreton et al., 2019). Several recent studies highlight the potential of oceanic islands to act as transitory repositories for plastic debris (Monteiro et al., 2018; Pham et al., 2020). As a result, it is unknown how much plastic is stored on coastlines in the IO, as well as how permanent this sink is." |
| | **Section 3 Observations** | |
| 66 | Perhaps this topic is the most approximated to a methodological topic. Thus, this topic should be worked on to improve the mechanism to search articles in this manuscript. Here, it could be defined the kind of reservoirs (biota - seabirds, invertebrates, mammals, reptiles; sediment – sand mud, water; deep sea) among many other variables. | We are unclear exactly what the reviewer means with this comment. |
| 67 | On the Scopus base, I used the following keywords "Indian Ocean" and plastic or microplastic and, I found 227 documents (1972-2021). After It was limited to review papers and, I found only seven articles and no one of them was about the purpose of this manuscript. Therefore, the authors have a great and fantastic study proposal. However, it needs to be improved, mainly in the methodology. After that, this article could be reference in plastic pollution in Indian Ocean. | Thank you. |
| | **Section 4** | |
| 68 | About the whole physical section (section 4)
The proposition of information among these sections with the other is too much different. The topics need an equilibrium because a review is constructed by a global vision of the theme. As a researcher, I know we ended up talking more about what we understand, but we have to control it. | We have shortened section 4 and completely removed section 5, see our response to comment #15. This should bring some more balance to the different parts of the paper. |
| | **Section 5** | |
| 69 | This topic is good writing in this manuscript because the authors bring a diversity of articles cited. But, because it is the Indian Ocean I think the manuscript should have more information about biological sinks since the literature has some articles about them.

Some suggestions of references:
- Cliff, G., Dudley, S. F., Ryan, P. G., Singleton, N., 2002. Large sharks and plastic debris in KwaZulu-Natal, South Africa. Marine and Freshwater Research, 53(2), 575-581. DOI: 10.1071/MF01146 | We have now removed section 5 from the manuscript, see our response to comment #15. We have added section 4.3.2 about ingestion as a possible (temporary) sink of plastics:

"Ingestion of plastics can occur at the ocean surface, in the water column, and on the |

- Carey, M. J. (2011). Intergenerational transfer of plastic debris by Short-tailed Shearwaters (Ardenna tenuirostris). Emu-Austral Ornithology, 111(3), 229-234.
- Roman, L., Paterson, H., Townsend, K. A., Wilcox, C., Hardesty, B. D., & Hindell, M. A. (2019). Size of marine debris items ingested and retained by petrels. Marine pollution bulletin, 142, 569-575.
- Ryan, P. G. (2008). Seabirds indicate decreases in plastic pellet litter in the Atlantic and south-western Indian Ocean. Mar. Pollut. Bull, 56, 1406-1409.
- Sparks, C., Immelman, S., 2020. Microplastics in offshore fish from the Agulhas Bank, South Africa. Marine pollution bulletin, 156, 111216. DOI: 10.1016/j.marpolbul.2020.111216
- Cartraud, A.E., Le Corre, M., Turquet, J., Tourmetz, J., 2019. Plastic ingestion in seabirds of the western Indian Ocean. Marine pollution bulletin, 140, 308-314. DOI: 10.1016/j.marpolbul.2019.01.065
- Crutchett, T., Paterson, H., Ford, B.M., Speldewinde, P., 2020. Plastic Ingestion in Sardines (Sardinops sagax) From Frenchman Bay, Western Australia, Highlights a Problem in a Ubiquitous Fish. Frontiers in Marine Science, 7, 526. DOI: 10.3389/fmars.2020.00526
- McGregor, S., Strydom, N.A., 2020. Feeding ecology and microplastic ingestion in Chelon richardsonii (Mugilidae) associated with surf diatom Anaulus australis accumulations in a warm temperate South African surf zone. Marine Pollution Bulletin, 158, 111430. DOI: 10.1016/j.marpolbul.2020.111430
- Hoarau, L., Ainley, L., Jean, C., Ciccione, S., 2014. Ingestion and defecation of marine debris by loggerhead sea turtles, Caretta caretta, from by-catches in the South-West Indian Ocean. Marine Pollution Bulletin, 84(1-2), 90-96. DOI: 10.1016/j.marpolbul.2014.05.031
- Pfeiffer, M. B., Venter, J. A., & Downs, C. T. (2017). Observations of microtrash ingestion in Cape vultures in the Eastern Cape, South Africa. African Zoology, 52(1), 65-67.
- Lavers, J. L., & Bond, A. L. (2016). Selectivity of flesh-footed shearwaters for plastic colour: evidence for differential provisioning in adults and fledglings. Marine environmental research, 113, 1-6.
- Cherel, Y., Xavier, J.C., de Grissac, S., Trouvé, C., Weimerskirch, H., 2017. Feeding ecology, isotopic niche, and ingestion of fishery-related items of the wandering albatross Diomedea exulans at Kerguelen and Crozet Islands. Marine Ecology Progress Series, 565, 197-215. DOI: 10.3354/meps11994

seafloor. Estimates of plastic ingestion by vertebrates (van Franeker, 2011; Davison and Ash, 2011), indicate that the global ingestion of plastics could be on the same order of magnitude as the amount of plastics accumulating in subtropical garbage patches (van Sebille et al., 2015). However, plastic ingestion is generally considered only a temporary and not a permanent sink of marine plastic debris.

Throughout the IO (Figure 2b), multiple studies have sampled ingested plastics in a variety of different fauna: benthic invertebrates (Taylor et al., 2016; Naidu et al., 2018), sessile invertebrates (Thushari et al., 2017), fishes (Ismail et al., 2018; Karthik et al., 2018; Baalkhuyar et al., 2019; Crutchett et al., 2020; McGregory and Strydom, 2020; Sparks et al., 2020), including large sharks (Cliff et al., 2002), seabirds (Cherel et al., 2017; Cartraud et al., 2019), turtles (Hoaru et al., 2014), bivalves (Naidu, 2019), and corals (Saliu et al., 2019). Recorded ingestion rates varied widely between species, from only approximately 0.4% of large sharks sampled (Cliff et al., 2002) to up to 90% of fish sampled (Sparks et al., 2020).

These sampling studies are both relatively few and relatively recent, so no estimates can be given about the total amount of plastic ingested by marine fauna in the IO, or about any trends in plastic ingestion. Cherel et al. (2017) did find that the

| | | wandering albatross chicks they investigated at Crozet and Kerguelen Islands had ingested low plastic loads compared to albatross chicks in the North Pacific Ocean. Crutchett et al. (2020) found low plastic ingestion levels in sardines compared to global levels. They also suggested that sampling plastics in globally common fishes, such as sardines, is a good way to compare and monitor ingestion rates between different locations around the world." |
| | | We have also added most of the suggested references to Table 1 (except a few that were not relevant for the IO or discussed land birds). |
| | | We have also added a section on the impact of plastic ingestion (section 5.2). |
| 70 | Please consider the information and suggestions given for improving the article. I think it has a lot of potentials, but we need to improve some points. The work done was a lot and I'm sure it can get even better. | Thank you – we have addressed the majority of your comments. |

---

## Author Response (AR2)

Figures 1 and 6 have been updated in response to the referee comments